# Hedgehog is relayed through dynamic heparan sulfate interactions to shape its gradient

Fabian Gude[1,9], Jurij Froese[1,9], Dominique Manikowski[1,9], Daniele Di Iorio[1], Jean-Noël Grad[2,8], Seraphine Wegner [1], Daniel Hoffmann [2], Melissa Kennedy[3,4,5,6], Ralf P. Richter [3,4,5,6], Georg Steffes[7] ✉ & Kay Grobe [1] ✉

Cellular differentiation is directly determined by concentration gradients of morphogens. As a central model for gradient formation during development, Hedgehog (Hh) morphogens spread away from their source to direct growth and pattern formation in *Drosophila* wing and eye discs. What is not known is how extracellular Hh spread is achieved and how it translates into precise gradients. Here we show that two separate binding areas located on opposite sides of the Hh molecule can interact directly and simultaneously with two heparan sulfate (HS) chains to temporarily cross-link the chains. Mutated Hh lacking one fully functional binding site still binds HS but shows reduced HS cross-linking. This, in turn, impairs Hhs ability to switch between both chains in vitro and results in striking Hh gradient hypomorphs in vivo. The speed and propensity of direct Hh switching between HS therefore shapes the Hh gradient, revealing a scalable design principle in morphogen-patterned tissues.

Pattern formation in multicellular organisms involves the differential specification of cell fate by a number of highly conserved signaling pathways, including the Hedgehog (Hh) pathway[1]. Hh forms a concentration gradient and directly specifies the fate of each cell along this gradient[2], therefore acting as a prototypical morphogen[3]. Yet, the mechanistic mode of Hh spread from producing to receiving cells, which is essential for graded morphogen action, is not clear. Among the various possible modes by which morphogen spread could be achieved, it is proposed that Hhs transport on filopodia called cytonemes[4,5], on secreted vesicles called exosomes[6], or on lipoproteins[7]. Hh diffusion from producing cells to target also meets many of the constraints imposed by the need for gradient precision, scalability, and accuracy[8–11]. However, extracellular Hh diffusion as the sole mechanism of Hh gradient formation is difficult to envision because patterning of folded epithelia is impossible if the morphogen diffuses off the plane of epithelial cell layers into the extracellular space, as this would prevent most Hh molecules from finding their receptor, Patched (Ptc), at some distance on the same epithelium[12]. One solution to this problem is to convert three-dimensional (3D) Hh diffusion into one-dimensional (1D) or two-dimensional (2D) transport. This may be accomplished by reversible Hh association with non-signaling interactors at the single-layered epithelial surface (as recently discussed in ref. [11]).

Indeed, genetic studies have established that Hh transport requires abundant expression of strongly negatively charged sulfated sugar chains at the cell surface that represent one class of non-signaling interactors. These are called heparan sulfate (HS). Hh binding HS consists of repeating disaccharide units of glucuronic acid/iduronic

[1]From the Institute for Physiological Chemistry and Pathobiochemistry, University of Münster, 48149 Münster, Germany. [2]The Institute for Bioinformatics and Computational Biophysics, University of Duisburg-Essen, 45117 Essen, Germany. [3]The School of Biomedical Sciences, Faculty of Biological Sciences, University of Leeds, Leeds LS2 9JT, UK. [4]The School of Physics and Astronomy, Faculty of Engineering and Physical Sciences, University of Leeds, Leeds LS2 9JT, UK. [5]The Astbury Centre for Structural Molecular Biology, University of Leeds, Leeds LS2 9JT, UK. [6]The Bragg Centre for Materials Research, University of Leeds, Leeds LS2 9JT, UK. [7]The Institute of Neuro- and Behavioral Biology, University of Münster, 48149 Münster, Germany. [8]Present address: Institute for Computational Physics, Universität Stuttgart, Allmandring 3, 70569 Stuttgart, Germany. [9]These authors contributed equally: Fabian Gude, Jurij Froese, Dominique Manikowski. ✉e-mail: steff_00@uni-muenster.de; kgrobe@uni-muenster.de

acid and N-acetylglucosamine with different degrees of sulfation[13] that can be linked to glycosyl-phosphatidyl-inositol (GPI)-anchored extracellular core proteins called glypicans (Glps). HS chains vary in size from ~5 to ~70 kDa, which translates into sufficient lengths of the unbranched polymers to cross a full-thickness basement membrane and touch an adjacent cell at distances of up to a hundred nanometers[14]. The essential role of these polyanionic non-signaling interactors in Hh transport was most convincingly demonstrated by mosaic analyses in the *Drosophila* wing disc, revealing that Hh did not cross even a single mutant cell deficient in HS biosynthesis or undergo incomplete sulfation in the gradient-forming field[15]. Consistent with this, embryonic null mutants for genes encoding HS-modified Glp core proteins[15–19] and HS biosynthetic enzymes (*ttv*[20,21] or *brother of ttv* and *sister of ttv*[22–24]) showed phenotypes similar to *hh* null mutants. These observations were previously explained by indirect HS functions in the binding of Hh/lipophorin complexes or the guidance of Hh-transporting cytonemes, by Hh stabilization in the matrix, or by the inhibition of Wingless (Wg) function that also contributes to parasegmental patterning of the *Drosophila* embryo[5,7,21]. In our work, we tested the alternative hypothesis that HS/Hh interactions mediate Hh spread directly. We also aimed to define the mechanistic requirements for such HS-directed spread and to resolve the paradox that Hh binding to cell surface-associated HS expands the Hh gradient[15] instead of restricting it—as would be expected from any mode of protein diffusion that is repeatedly interrupted by temporal protein immobilization at the cell surface.

To this end, we modulated direct interactions of *Drosophila melanogaster* Hh with HS and heparin (a highly sulfated form of HS) by site-directed mutagenesis of the Hh ligand. It is known that Hh binds HS/heparin via two independent binding sites: the Cardin-Weintraub (CW) motif as part of the N-terminal extended Hh peptide[25–27] and a second motif containing arginines (R) 238 and 239 as part of the *Drosophila* Hh globular domain (corresponding to lysine (K) 178 in vertebrate Sonic Hh (Shh))[28,29]. Our analyses focused on the second motif because CW modifications at the N-terminus would change several other important aspects of Hh and Shh biology in vivo as well, including protein clustering[30], Hh bioactivity[31], or solubilization[32,33]. We exchanged the basic arginines 238 and 239 of the globular HS-binding site for neutral alanines (Hh$^{R/A}$) or acidic glutamates (Hh$^{R/E}$) and found that Hh$^{R/A}$ and Hh$^{R/E}$ variant expression under endogenous promotor control from the *Drosophila* genomic sequence[34] generated Hh gradient hypomorphs in tissues that require morphogen spread over a relatively long-range (i.e., tens of micrometers[35]), but less so in tissues that depend on Hh transport over a shorter range (i.e., up to a few micrometers). In accordance with our hypothesis that Hh-HS interactions rely on ionic interactions, exchanging the positively charged arginines to neutral alanines resulted in milder hypomorphic phenotypes than did exchange arginines to negatively charged glutamic acids. We also found that interfering with ionic HS interactions of vertebrate Hh$^{R/A}$ and Hh$^{R/E}$ orthologs in vitro strongly impaired their ability to simultaneously bind and cross-link two individual HS chains and to quickly and directly traverse between them. Together, these findings reveal that HS-dependent Hh spread depends on a set propensity and speed of direct Hh switching between HS chains to control the Hh gradient over longer distances (which involves multiple switching events) more than Hh spread over shorter distances (which requires fewer HS-switching events). This model also explains the inability of Hh to cross fields of cells deficient in HS production in vivo[15], as described previously, and the similarity between *hh* null mutant embryonic phenotypes and phenotypes of null mutants for genes encoding HS or Glps[15–24]. We suggest that the design principles of HS-mediated Hh spread may be broadly applicable to other HS-binding morphogens, as well as to the spread of oligomeric HS-binding inflammatory chemokines[36].

## Results

### Two heparin interaction sites in vertebrate Shh and modeled *Drosophila* Hh

Hh/HS ionic interactions are mediated by the binding of positively charged lysine and arginine residues on Hh to sulfated, negatively charged motifs expressed on linear HS chains. In the structure of the vertebrate Hh family member Shh (Protein Data Bank (PDB): 3m1n, Fig. 1a, b), two positively charged clusters of K/R residues were previously defined: the N-terminal CW site consisting of amino acids K32, R33, R34, K37, and K38 (human Shh nomenclature)[25] and a highly conserved second site consisting of basic amino acids K45, K87, R153, R155, and K178[28,37] (Supplementary Fig. 1). In *Drosophila* Hh structures obtained by homology modeling from the 3m1n Shh structure, these sites correspond to CW amino acids R93, R95, and R97 and to amino acids K105, R147, R213, R238, and R239 of the second site, respectively (Fig. 1c, d and Supplementary Fig. 1). Analysis of molecular dynamics (MD) simulations of *Drosophila* Hh structures shows that both motifs are far away from each other in the monomeric state of Hh, with a distance of ~2.7 nm. This distance was not significantly different between 3m1n and the two Hh structures obtained by homology modeling, Hh model M and Hh model S (Supplementary Tables S1, S2). This suggests that, in addition to the possibility that both sites cooperate in *Drosophila* Hh binding to only one HS chain, Hh may bind to two individual HS chains simultaneously.

To further predict the relative contributions of both sites for Hh/HS interactions, we screened Hh and Hh variants lacking one or both fully functional motifs with a heparin disaccharide as a charged ligand in silico using a rigid body correlation method[38]. The calculated electrostatic contribution to the change in binding energies ($\Delta\Delta G^{elec}$) between wild-types and mutants were then used as a score to rank the Hh mutants against each other. We limited mutagenesis of the second site (in Hh$^{R/A}$) to R238/R239 because basic Hh amino acids K105, R147, and R213 correspond to Shh amino acids K45, K87, and R153, which play an additional role in Shh binding to Ptc (Fig. 1b, d, violet)[37,39]. On the basis of these calculations, the combined mutagenesis of both binding sites (in Hh$^{R/A;CW/A}$) is predicted to strongly impair the heparin binding of the fly morphogen (Fig. 1e and Supplementary Table 3). Mutation of CW residues R93/95/97 to alanines (in Hh$^{CW/A}$) is in silico predicted to still allow for heparin-binding because of the second binding site present in the globular domain, and the Hh$^{R/A}$ or Hh$^{R/E}$ mutants lacking R238/239 are also predicted to bind because of the N-terminal CW motif. This suggests that both Hh motifs contribute to HS-binding, but that neither of the two sites is absolutely essential for HS-binding. Moreover, in both Shh and Hh, mutants with glutamates in their second binding sites have a lower affinity to heparin than mutants with alanines (Fig. 1e and Supplementary Table 3). We then tested whether R238/239 mutagenesis affected other aspects of Hh$^{R/A}$ biofunction in vitro. To this end, we expressed wild-type and mutated proteins in *Drosophila*-derived S2 cells and confirmed their unimpaired autoprocessing, secretion, and closely correlated protein stability in solution (Fig. 1f and Supplementary Fig. 2). Moreover, Hh, Hh$^{R/A}$, and Hh$^{R/E}$, or Shh, Shh$^{K/A}$ and Shh$^{K/E}$ showed similar bioactivities in a cell-based in vitro differentiation assay that is specific for Hh biofunction (Fig. 1g, h). This indicated that K178 and corresponding Hh amino acids R238/R239 are not essential for interactions with the Ptc receptor, in full accordance with recent structural findings[39].

### Modification of the endogenous *hh* DNA locus in vivo

To determine the role of HS-binding Hh amino acids R238/239 in vivo, we used a previously published strategy to replace the endogenous *Drosophila melanogaster hh* locus with hh, hh$^{R/A}$ or hh$^{R/E}$ cDNA for their in vivo expression under endogenous promotor control[34] (Supplementary Fig. 3). To this end, we cloned all transgenes into the reintegration vector (RIV$_{white}$) to target them into hh[KO] flies that had the first exon of endogenous *hh* replaced and an *attP* landing site inserted

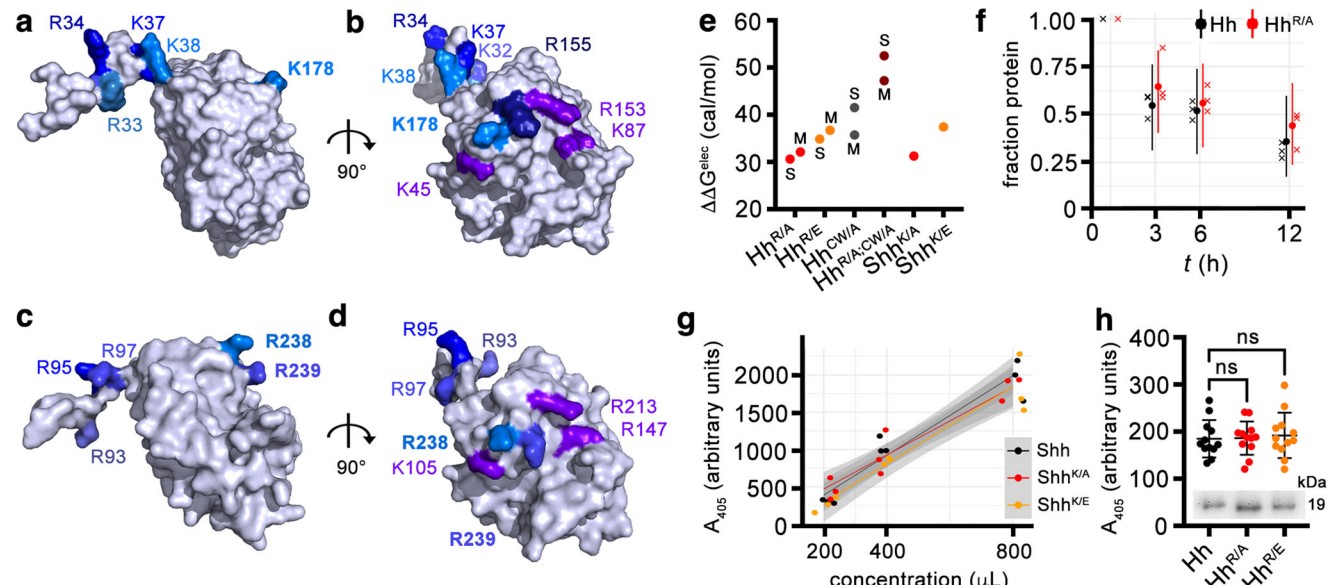

**Fig. 1 | Two HS-binding sites in Shh and modeled *Drosophila melanogaster* Hh.**
**a**–**d** The first established HS-binding site (the Cardin-Weintraub (CW) motif) of Shh (**a**, **b**, pdb: 3m1n) or of *D. melanogaster* Hh (**c**, **d**) is located at the N-terminal extension of one monomer (left in **a**, **c** and top in **b**, **d**), whereas the second site comprises residues on the globular domain. HS-binding basic residues are in different shades of blue and residues that also bind to Ptc receptors are in violet. Both HS-binding sites are far from each other in the monomeric structures, with a median distance of ~2.7 nm. See Supplementary Tables 1, 2 for details. **e** In silico analysis of the effect of mutations in wild-type *Drosophila* HhNΔ251–257 and human ShhNΔ191–197 on the electrostatic contribution to the binding affinity against a heparin disaccharide. ΔΔG$^{elec}$ values express the difference of ΔG$^{elec}$ between each mutant protein and its corresponding wild-type protein. In Hh, both the CW domain and globular domain contribute to heparin binding. In both Shh and Hh, the globular glutamate variants have a lower affinity to heparin than the alanine variants. Annotations M and S distinguish results based on different template structures of protein (Electrostatic calculations in Methods, Source data are provided as a Source Data file). **f** Remaining fraction of Hh (black) and Hh$^{R/A}$ (red) after 0, 3, 6, and 12 h. Crosses correspond to measured data with small shifts along the time axis used to separate points for clarity. Dots and uncertainty bar mark mean fractions estimated with a Bayesian model (see supplementary information), with the dot marking the median estimate and the uncertainty bars covering 5 to 95% quantiles. As the uncertainty bars have large overlaps, we cannot distinguish the decay dynamics of Hh and Hh$^{R/A}$ with confidence. Raw data and 5 to 95% quantiles are provided as Source Data file. **g** A$_{405}$ measurements for each of the three protein variants (color legend) at three different concentrations are shown as dots. A random jitter in the horizontal direction was added to avoid overplotting. The three lines correspond to linear models fitted with the data of the respective proteins. 95%-confidence ribbons (gray areas) for the linear models largely overlap, indicating that courses of data for different proteins cannot be distinguished with confidence. Raw data and 95%-confidence ribbons are provided as Source Data file. **h** Hh, Hh$^{R/A}$, and Hh$^{R/E}$ induce Hh-specific differentiation of C3H10T1/2 reporter cells to similar degrees, confirming that the arginines do not contribute to Ptc receptor binding. Inset: similar amounts of Hh variants were used in this experiment. One-way ANOVA, Dunnett's multiple comparisons test, $n = 12$ for each set. Error bars represent standard deviation. $F = 0.098$. $p = 0.9$. Source data, means and $p$ values (n.s.: $p > 0.05$) are provided as Source Data file.

into the 5′ untranslated region (5′UTR). When homozygous, these modifications abolished all endogenous *hh* function[34]. These flies were also mutant in the endogenous *white* locus, allowing post-targeting detection of the *mini-white* of RIV$_{white}$ before its subsequent Cre-mediated removal (Supplementary Fig. 3). Accurate targeting of all constructs into the hh[KO] locus was confirmed by PCR and sequencing of the targeted loci (Supplementary Fig. 3). We obtained Hh[KO;hh]/hh[KO;hh] flies with both endogenous *hh* coding sequences replaced by hh cDNA at the expected Mendelian ratio, and flies were fertile, demonstrating that spatiotemporal expression, folding, secretion, transport, and binding of transgenic Hh to the receptor Ptc were not impaired. Rather, Hh biofunction seemed slightly increased because close examination revealed subtle eye and wing gain-of-function phenotypes (Fig. 2a, b). We continued to use these two developmental systems as easily and reliably quantifiable read-outs for two different mechanistic modes of Hh gradient formation and signaling.

In the *Drosophila* eye, photoreceptor cells develop in a wave of differentiation, called the morphogenetic furrow, that moves from the posterior to the anterior of the eye anlage (the eye-antennal disc). Cells located anterior to the furrow respond to Hh secreted by differentiated cells directly posterior to the furrow by eventually producing the same protein. This generates a cyclic Hh signal that progresses the furrow across the disc[40] and determines the number of photoreceptors (ommatidia) in the adult eye. For this reason, certain eye-promotor-

specific mutations within the *hh* gene involved in this cyclic process, such as the viable *hh$^{bar3}$* allele, cause fewer ommatidial columns, which are fewer still if in trans with *hh$^{AC}$* loss-of-function alleles[32]. We found that ommatidial numbers in hh[KO;hh]/hh[KO;hh] flies averaged 725 ± 33 ommatidia/eye, compared with 670 ± 20 ommatidia/eye in w$^{1118}$ and 664 ± 32 ommatidia/eye in heterozygous hh[KO;hh]/+ controls ($n = 10$ for each genotype, Fig. 2a). We suggest that the subtle gain-of-function phenotype of transgenic *hh* in vivo may be caused by attP/B insertion into the 5′UTR translational regulator (Supplementary Fig. 3). We also observed moderate Hh gain-of-function in the second *Drosophila* developmental system studied in our work, the wing. Here, Hh is expressed in the posterior compartment of the wing anlage, called the wing disc, and moves anteriorly at the folded epithelial surface to reach ~12 cell rows in the anterior wing disc compartment (summarized in refs. [41,42]). Therefore, in contrast to the eye disc where the morphogen source (the morphogenetic furrow) moves to form the compound eye, it is the morphogen that moves ~10 μm across the anterior wing disc compartment[35]. In adult wings, the position of the longitudinal L3/L4 veins (Fig. 2b, inset) results from this molecular movement and therefore is one read-out of the positional information provided by high Hh concentrations close to the posterior Hh source[41]. We found the ratio between L3/L4 and the L2/L3 intervein area anterior to the central L3/L4 domain to be increased from 1.11 ± 0.02 (w$^{1118}$) to 1.17 ± 0.02 (hh[KO;hh]/hh[KO;hh]), $p < 0.0001$, $n = 13$, and $n = 27$, respectively (Fig. 2b). This small yet significant gain-of-function

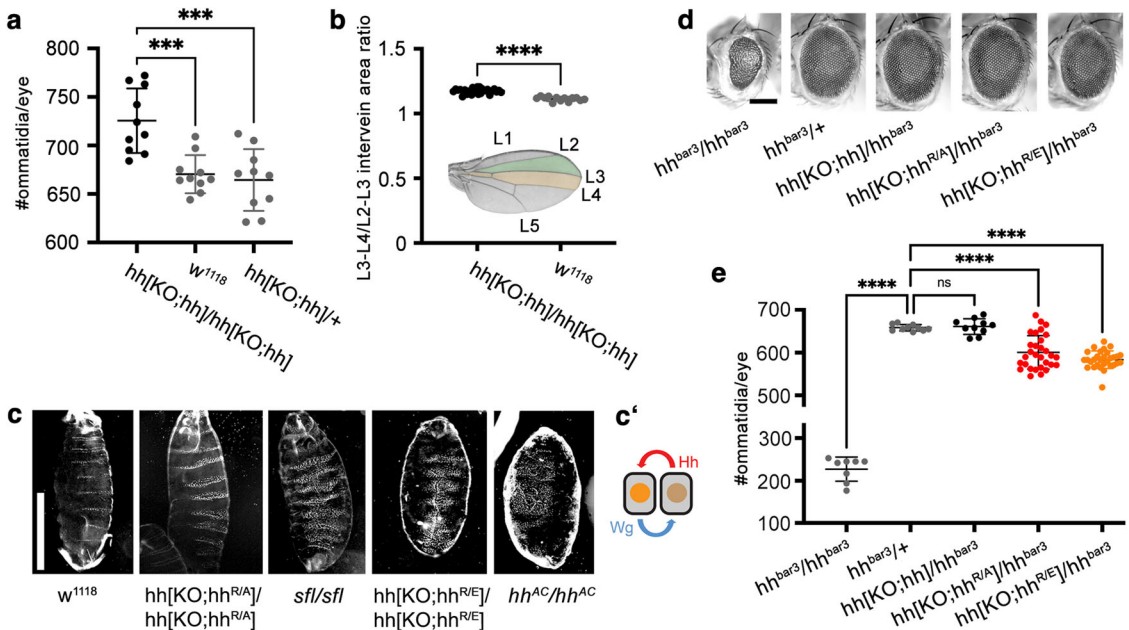

**Fig. 2 | Opposing phenotypes of flies expressing *hh* versus *hh*[R/A] or *hh*[R/E] under endogenous promotor control. a** Quantification of the numbers of ommatidia per eye in female flies. Hh biofunction in eye disks of flies homozygous for transgenic *hh* (in hh[KO;hh]/hh[KO;hh] flies) is increased over that in w[1118] or heterozygous controls. Statistical significance was determined by one-way ANOVA followed by Dunnett's multiple comparison test. $n = 10$ for each genotype, $F = 13.6$. Error bars represent standard deviation. ***$p \leq 0.0004$. Source data are provided as a Source Data file. **b** Inset: Hh produced in the posterior compartment of the wing disc directly patterns the L3-L4 intervein area (orange) and indirectly patterns the L2-L3 intervein area (green). Quantification of L3-L4/L2-L3 intervein areas revealed significantly increased patterning activities of transgenic Hh. Statistical significance was determined by unpaired *t*-test (two-tailed). $n = 13$ wings for w[1118] and $n = 27$ for hh[KO;hh]/hh[KO;hh], ****$p < 0.0001$. L1-5 indicate the five longitudinal *Drosophila* wing veins. Source data are provided as a Source Data file. **c** Segment phenotype changes in hh[KO;hh[R/A]]/hh[KO;hh[R/A]] larvae resemble those in *sfl/sfl* larvae that are

deficient in HS biosynthesis, and hh[KO;hh[R/E]]/hh[KO;hh[R/E]] larvae resemble *hh[AC]/hh[AC]* larvae that lack hh expression. Segment phenotypes were consistently observed in three independent crosses. Scale bar: 100 µm. **c′** Schematic of the self-reinforcing Hh/Wg signaling loop at one of the 14 parasegment boundaries in *Drosophila* embryos. Due to Hh/Wg interdependency to maintain the loop, depletion of either morphogen will result in embryos that lack a naked cuticle. **d** Transgenic *hh* and both *hh* variants restore eye development in flies homozygous for impaired endogenous *hh* expression specifically in the eye (*hh[bar3]/hh[bar3]*). This demonstrates the largely retained bioactivity of both Hh variants in this system. Scale bar: 100 µm. **e** Quantification of the numbers of ommatidia per eye of female flies as shown in (**d**). Statistical significance was determined by one-way ANOVA followed by Dunnett's multiple comparison test. $n = 8$ for hh[bar3]/hh[bar3], $n = 10$ for hh[bar3]/+ and hh[KO;hh]/hh[bar3], and $n = 30$ for hh[KO;hh[R/A]]/hh[bar3] or hh[KO;hh[R/E]]/hh[bar3]. Error bars represent standard deviation. $F = 357$. ****$p \leq 0.0001$, ns: $p = 0.99$. Source data are provided as a Source Data file.

confirmed that transgenic Hh, when expressed under endogenous promotor control, was fully functional in vivo.

## Arrested embryonic development in flies producing Hh[R/A] and Hh[R/E]

In contrast to hh[KO;hh]/hh[KO;hh] flies that express hh cDNA, homozygous hh[R/A] and hh[R/E] expression from the same locus resulted in segment phenotypes typical for embryos made deficient in hh expression (*hh[AC]*) or in the biosynthesis of HS (*sulfateless* (*sfl*) mutants with undersulfated HS or *tout-velu* (*ttv*) mutants that lack HS expression)[15,16,21–23] (Fig. 2c and Supplementary Fig. 4). All these embryos are characterized by more or less the complete absence of naked cuticle (which is visible as dark segmental regions in Fig. 2c, labeled "n" in Supplementary Fig. 4) and its replacement by denticle-covered cuticle (visible as white "stripes" in Fig. 2c, labeled "d" in Supplementary Fig. 4). In notable contrast, expression of hh[R/A] and hh[R/E] in trans with the eye-specific *hh[bar3]* allele revealed significant bioactivities of both gene variants during development of the *Drosophila* compound eye (Fig. 2d). Ommatidial numbers in *hh[bar3]*/+ and hh[KO;hh]/hh[bar3] flies averaged $658 \pm 7$ ommatidia/eye and $661 \pm 18$ ommatidia/eye, respectively, compared with $227 \pm 28$ ommatidia/eye in *hh[bar3]/hh[bar3]* eyes. Ommatidial numbers in hh[KO;hh[R/A]]/*hh[bar3]* flies averaged $601 \pm 39$ ommatidia/eye, and in hh[KO;hh[R/E]]/*hh[bar3]* flies averaged $584 \pm 20$ ommatidia/eye ($n = 10$ for *hh[bar3]/hh[bar3]*, *hh[bar3]*/+ and hh[KO;hh]/hh[bar3], $n = 30$ for hh[KO;hh[R/A]]/*hhbar3* and hh[KO;hh[R/E]]/*hh[bar3]*, female flies were analyzed, Fig. 2e). This demonstrates Hh

mutagenesis did not abolish morphogen production and the capacity to signal to adjacent target cells. It further demonstrates that deletion of HS-binding Hh amino acids R238/239 affects the reciprocal, self-repeating, and self-reinforcing Hh/Wingless signaling loop (Fig. 2c′) required for embryonic cell specification at the parasegment boundary[43,44] to a much stronger degree than cell-to-cell signaling in the eye disc[40].

## Hh[R/E] expression in the eye-antennal disc selectively impairs Hh signaling

As described previously, in the *Drosophila* compound eye, the source of Hh production itself – the morphogenetic furrow – moves from the posterior to the anterior of the developing retina at constant speed[40] to drive cyclic Hh signaling and photoreceptor differentiation across the eye primordium (Fig. 3a, cartoon). As shown in Fig. 2e, hh[R/A] and hh[R/E] alleles in trans with *hh[bar3]* allowed this process to occur. To complement this finding, we next aimed to analyze cyclic signaling over short distances and furrow progression in hh[KO] eye disks from cell clones that lack *hh* expression or that express hh[R/A] or hh[R/E] from both alleles. However, one known problem of this approach is that the mechanisms of cell competition in proliferating epithelia (such as imaginal disks) eliminate small "aberrant" clones that deviate in properties from their surrounding neighbors[45–47], and that organ size is subsequently restored to wild-type level at the expense of the mutant of interest. We found that this was also true for hh[KO] clones induced by mitotic recombination in eye and wing with standard clonal/mitotic

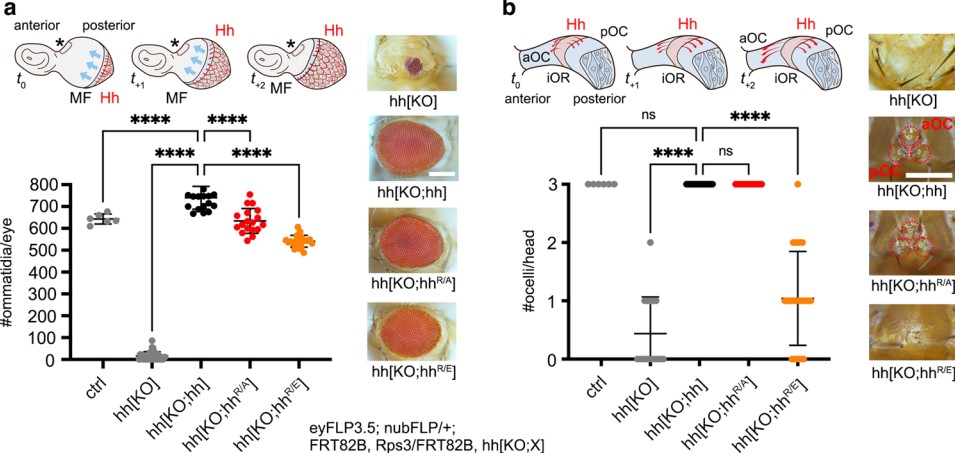

**Fig. 3 | Eye-disc-specific hh variant expression from clones in moving and static sources. a** The cartoon shows reiterated cell-to-cell Hh signaling from a clonal source (Supplementary Fig. 5) moving at a particular pace to drive photoreceptor differentiation across the eye primordium. Red arrows denote Hh spread, blue arrows denote the movement of the morphogenetic furrow (MF), $t_0$-$t_{+2}$ indicates early and later stages. The asterisk and dashed area indicate the ocellar field (as schematically shown in **b**). Clones homozygous for hh[KO] do not express *hh*, which severely impairs eye development, but hh or hh variant cDNA expression from both alleles restores eye development. Scale bar: 100 μm. Despite the observed restoration of eye development, quantification of ommatidia numbers per eye of female flies reveals significantly impaired activities of both hh variants. Statistical significance was determined by one-way ANOVA followed by Dunnett's multiple comparison test. Error bars represent standard deviation. $n = 6$ (hh), $n = 32$ (hh[KO]), $n = 19$ (hh[KO;hh]), $n = 18$ (hh[KO;hh$^{R/A}$]), and $n = 22$ (hh[KO;hh$^{R/E}$]), $F = 1474$, ****$p \leq 0.0001$. Ctrl: FRT82B donor line. Source data are provided as a Source Data file. **b** More severely affected development of another eye disc area (as shown in the cartoon adapted from[52]) that uses the same Hh signaling components,

but differs from the system described in (**a**) in that it depends on Hh signaling over a relatively long distance (tens of micrometers[52]) from a static source (indicated by red bent arrows). The asterisk in (**a**) marks this ocellar region in the disc. Clones homozygous for hh[KO] do not express *hh* in this area and strongly impair the development of one or more of the three camera-type eyes called anterior and posterior ocelli (aOC and pOC, red circle, the *hh* expression domain will become the interocellar region (iOR), Supplementary Fig. 5). Clones homozygous for hh[KO;hh$^{R/E}$] also impaired their development. The average number of ocelli per hh[KO] fly: $0.44 \pm 0.6$ ocelli, $n = 16$. hh[KO;hh$^{R/E}$]: $1 \pm 0.8$ ocelli, $n = 24$. The normal number of three ocelli/fly was observed for all other genotypes. Error bars represent standard deviation. No sex bias was observed in any assay. Statistical significance was determined by one-way ANOVA followed by Dunnett's multiple comparison test, $F = 99$. $n = 6$ (hh), $n = 16$ (hh[KO]), $n = 20$ (hh[KO;hh]), $n = 18$ (hh[KO;hh$^{R/A}$]), and $n = 24$ (hh[KO;hh$^{R/E}$]). ****$p \leq 0.0001$, ns: $p > 0.99$. Scale bar: 100 μm. Source data are provided as a Source Data file. Bottom: Fly genotype used in these experiments, the X denotes no hh (hh[KO]), hh (hh[KO;hh]), or hh variants.

recombination using eyFLP3.5 and nubFLP with or without MARCM[48], because this resulted in fully restored wildtype eyes and wings by the proliferation of the surviving twin spot clones and potentially residual heterozygous cells. We solved this problem by including the "Minute" mutation Rps3 (Ribosomal protein S3 that encodes a component of the small subunit of cytoplasmic ribosomes) on chromosomal arm 3 R to generate a meaningful and quantifiable hh[KO] baseline phenotype. Because the dominant Rps3 mutation limits ribosomal capacity and protein biosynthesis, cells heterozygous for the Rps3 mutation have a competitive disadvantage, and cell clones homozygous for Rps3 die after the induction of mitotic recombination and are eliminated from the tissue. As a consequence, the organ within the clonal area is formed from homozygous cells that lack Rps3 mutations and are not eliminated from the tissue[47–49], allowing us to analyze morphogenetic gradients build exclusively by the endogenously controlled alleles of interest (Supplementary Fig. 5 and Supplementary Movie 1). We confirmed this technique using flies with clones homozygous for the hh[KO] null allele upon eyFLP3.5-mediated mitotic recombination. This almost completely abolished eye development ($16 \pm 20$ ommatidia/eye, $n = 32$, 11 flies without ommatidia, Fig. 3a). As a positive control, flies with clones homozygous for hh[KO;hh] in the eye disc that express transgenic hh under endogenous promotor control restored ommatidia numbers in adult compound eyes ($740 \pm 52$ ommatidia/eye, $n = 19$, Fig. 3a). Ommatidial numbers in flies with eye disc clones homozygous for hh[KO;hh$^{R/A}$] averaged $634 \pm 56$ ommatidia/eye ($n = 18$) and $541 \pm 27$ ommatidia/eye if derived from eye disc clones homozygous for hh[KO;hh$^{R/E}$] ($n = 22$, Fig. 3a). This shows that both Hh variants are bioactive and that compound eye development was restored to significant degrees, but with Hh$^{R/E}$ being less potent than Hh$^{R/A}$. This is consistent with negatively charged (HS-repulsive) amino

acids in Hh$^{R/E}$ and the presence of neutral amino acids in Hh$^{R/A}$ that is therefore expected to bind HS to a stronger degree.

In *Drosophila*, the mono-layered eye-antennal disc also gives rise to a second component of the fly visual system called the ocellar complex. This complex comprises one small anterior camera-type eye, the anterior ocellus (aOC), and two posterior ocelli (pOC) located on the forehead of the fly (Fig. 3b and Supplementary Fig. 5). Similar to the compound eye, Hh signaling specifies the ocellar complex[49–51], with the important difference that the prospective ocellar complex region of the disc is formed by a bilaterally signaling domain of Hh expression flanked by two regions that are competent to differentiate into ocellar photoreceptors (Fig. 3b, cartoon and Supplementary Fig. 5). As a consequence, ocellar specification results from a "wave" of Hh molecules with sufficient speed to spread over the entire field of both ocellar competence domains[52], reaching cells up to 40 μm (~10 cells) from the Hh source. Similar to what we observed in the compound eye, ocellar development upon eyFLP3.5-mediated hh[KO] mitotic recombination was strongly disrupted (no ocelli formed in ten flies, one ocellus in five flies, and two ocelli in one fly), whereas homozygous clonal hh[KO;hh] and hh[KO;hh$^{R/A}$] expression always resulted in the formation of three ocelli (Fig. 3b). In notable contrast, ocelli formation in disks with clones homozygous for hh[KO;hh$^{R/E}$] was significantly impaired, resulting in six flies with no ocelli, 12 flies with one ocellus, five flies with two ocelli, and only one fly with three (smaller) ocelli. Together, these findings suggest that HS-binding Hh amino acids R238/239 contribute moderately to cyclic Hh signaling to relatively close-by target cells (i.e., within micrometers) in regions of the eye disc that give rise to the compound eye, but may contribute more significantly to a Hh signal sustained from a laterally restricted source (i.e., within tens of micrometers) in the same tissue[52].

## Hh<sup>R/A</sup> and Hh<sup>R/E</sup> expression impair the Hh signaling range in the wing disc

In the wing, Hh is expressed in the L3 larval posterior compartment and moves anteriorly to reach ~12 cell rows in the anterior compartment[35,41,42]. This anterior movement can be read out by well-established markers, or more easily in the imago by the L3-L4 intervein field area (Fig. 4a). To determine whether HS-binding Hh amino acids R238/239 contribute to spatial patterning in the wing disc, we generated large clones that endogenously express homozygous hh, hh[R/A], or hh[R/E] by nubbin (nub)-FLP-mediated mitotic recombination, and analyzed morphogenetic gradients build exclusively by the endogenously controlled alleles of interest (Supplementary Fig. 5). Quantification of the ratio between L3-L4/L2-L3 intervein areas served as one established read-out for Hh signaling range. We found L3-L4/L2-L3 intervein area ratios ranged from $0.47 \pm 0.26$ (upon homozygous clonal hh[KO] expression, $n = 26$) to 1.10 ± 0.03 (hh[KO;hh], $n = 39$) at 25 °C. L3-L4/L2-L3 intervein area ratios resulting from wing disc clones homozygous for hh[KO;hh<sup>R/A</sup>] were reduced (1.0 ± 0.04, $p \leq 0.002$, $n = 34$), and those from clones expressing hh[KO;hh<sup>R/E</sup>] were even smaller (0.69 ± 0.09, $p \leq 0.0001$, $n = 43$) (Fig. 4a, b). This indicates that HS-binding Hh amino acids R238/239 also contribute charge-dependently to the Hh signaling range sustained from the posterior into the distant anterior compartment of the *Drosophila* wing disc.

## Hh<sup>R/A</sup> and Hh<sup>R/E</sup> "relay" between HS chains is impaired

How can the important role of Hh amino acids R238/239 in the patterning of the wing and part of the eye disks—that both require morphogen spread over relatively long distances of tens of micrometers[35,52]—be explained? It is established that Hh binds cell surface HS on producing wing disc cells prior to its release[30] and that continuous HS expression in the anterior gradient-forming field facilitates Hh transport in vitro[53] as well as in the disc[15]. From these observations, we hypothesized that Hh may remain associated with the epithelial surface through its affinity for HS, and that the role of the second HS-binding area may be to bind an adjacent HS chain as a prerequisite to directly switch between two chains to facilitate Hh transport.

We tested this hypothesis by quartz crystal microbalance with dissipation monitoring (QCM-D). The core of the QCM technology is an oscillating quartz crystal sensor disc with a resonance frequency related to the mass of the disk. This allows the real-time detection of nanoscale mass changes on the sensor surface by monitoring changes in the resonance frequency ($\Delta F$): adsorption of molecules to the surface decrease $F$, whereas a mass decrease will increase $F$. QCM-D measures an additional parameter, the change in energy dissipation $D$, which is particularly useful in the study of soft layer properties, with $\Delta D$ correlating with the layer softness. Using this technique, we analyzed Shh instead of *Drosophila* Hh because the latter proved difficult to produce recombinantly and purify—possibly due to the lack of the stabilizing zinc ion in Drosophila Hh[54]. Shh interaction surfaces were built on fluid-supported lipid bilayers (SLBs), by adhesion and rupture of small unilamellar vesicles (SUVs) containing 5 mol% biotinylated lipids, as a plasma membrane model with streptavidin monolayer coating. Highly sulfated heparin was anchored via a biotin moiety at the reducing end to the streptavidin, and served as a proxy for cell surface HS (Fig. 5a' and Supplementary Fig. 6)[55,56]. Like cell surface HS attached to GPI-linked Glps, streptavidin-linked heparin can rotate freely and move laterally on the sensor surface[55]. We then added the protein-ligand and monitored nanoscale mass changes upon protein adsorption to the heparin surface, as indicated by decreased resonance frequencies ($-\Delta F$) of the sensor, as well as changes in the softness of the heparin film upon protein binding (as reflected in the dissipation shift ($\Delta D$)). As shown in Fig. 5a, incubation of the QCM-D chip with thrombin, a prototypic nonspecific HS-binding protein with one HS interaction site[57], induced a frequency shift ($-\Delta F$) of about

−40 Hz, indicating effective binding, and proportionally increased energy dissipation ($\Delta D$), due to a relatively soft heparin/thrombin film. Both parameters were reversed after washing buffer injection into the QCM-D chambers, indicating fast thrombin wash-off from heparin (Fig. 5a, a").

The addition of purified unlabeled Shh to the QCM-D chip induced a much faster frequency decrease to −40 Hz (despite a five times lower concentration) compared to thrombin, indicating faster binding kinetics, and most proteins remained bound during injection of the wash buffer (Figs. 5b, b', 6a). Most importantly, and in stark contrast with thrombin, we observed that the $\Delta D$ of the layer decreased proportionally during Shh binding, suggesting protein-induced stiffening of the heparin film. Analogous assays with two distinct HS samples instead of heparin confirmed that Shh binding with associated HS film rigidification are qualitatively preserved over a range of HS sulfation levels (Supplementary Fig. 6b). Moreover, complementary assays by fluorescence recovery after photobleaching (FRAP) demonstrated that protein binding reduced the lateral diffusion of HS chains (Supplementary Fig. 6c). Together, the increased rigidity (evidenced by QCM-D) and reduced HS mobility (evidenced by FRAP) indicate that Shh effectively cross-links HS/heparin chains, consistent with the presence of two Shh binding sites for HS/heparin[28] (Figs. 5b', 6b).

Heparin interactions with unlabeled Shh<sup>K/E</sup> variants lacking HS-binding K178 (Fig. 1a, b) were reduced (Fig. 6a–c), as indicated by their decreased binding rate, and their increased unbinding rate during the buffer A wash (Fig. 6d). We conclude that the globular HS-binding site of Shh, and K178 that is part of this site, makes a sizeable contribution to the affinity ($K_D = k_{off}/k_{on}$) of the protein/heparin interaction: K178 replacement increased the concentration of half-maximal binding $K_{0.5}$ (as detected by QCM-D) from 26.5 ± 2.6 nM (Shh) to 83.6 ± 13.6 nM (Shh<sup>K/E</sup>, Supplementary Fig. 7a). Heparin interactions with unlabeled Shh<sup>K/A</sup> were less affected (Fig. 6a, b, d and Supplementary Fig. 7a), in line with in silico predictions (Fig. 1e). We also found that heparin film rigidification (as a consequence of Shh-mediated heparin cross-linking) was reduced (Shh<sup>K/A</sup>) or effectively abolished (Shh<sup>K/E</sup>) compared to the wild-type protein, as demonstrated by a lesser decrease and an increase, respectively, in $\Delta D$ (Fig. 6a–c, e). The increase in $\Delta D$ for Shh<sup>K/E</sup> is analogous to thrombin with one prototypical HS-binding site (Fig. 6a), suggesting that the second binding site in Shh<sup>K/E</sup> is effectively impaired. The strongly reduced cross-linking capacity of Shh<sup>K/E</sup> was independently confirmed by FRAP. Here, Shh strongly reduced the capacity of fluorescently labeled, streptavidin-coupled heparin to move laterally on supported lipid bilayers, and incubation with Shh<sup>K/E</sup> increased recovery speed (Supplementary Fig. 7b, c).

Having established that Shh but not Shh<sup>K/E</sup> were able to cross-link HS, we next tested if the capacity of Shh to simultaneously bind two HS chains facilitates direct morphogen switching between the chains. To this end, we injected soluble heparin onto the Shh-loaded heparin film. As shown in Fig. 6a, soluble heparin indeed served as a very effective acceptor, rapidly eluting most Shh from the QCM-D sensor surface. In stark contrast, we found that the propensity of Shh<sup>K/A</sup>, and even more so Shh<sup>K/E</sup>, to switch to soluble heparin was strongly reduced and the time required to switch strongly increased (indicated by reduced relative $\Delta F$ and increased $\Delta t$; Fig. 6a–c, f). These observations provide a direct link between the simultaneous engagement of two HS chains and facilitated switching between the chains. Finally, we tested the capacity of Shh to switch between HS chains of different sulfation and charge; i.e. we hypothesized that Shh moves between chains of equal overall charge, but may not switch from higher sulfated HS to lower sulfated (e.g., less charged) chains. To this end, we injected equal amounts of selectively de-N-sulfated soluble heparins (that lack the exact sulfates that require *sfl* activity during biosynthesis), de-6O-sulfated soluble heparin, and de-2O-sulfated soluble heparin before the final heparin injection into the QCM-D chambers. As shown in Fig. 7a,

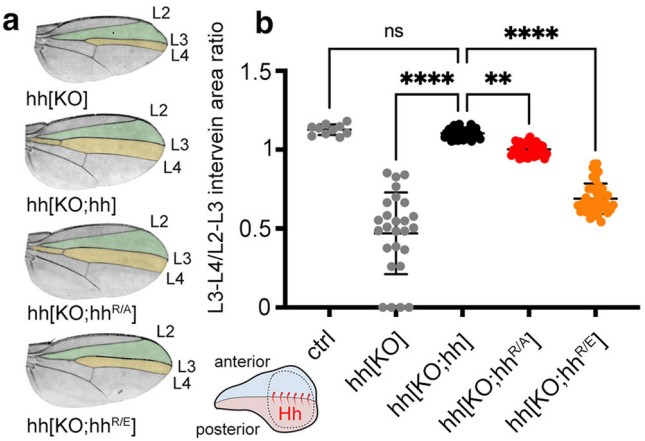

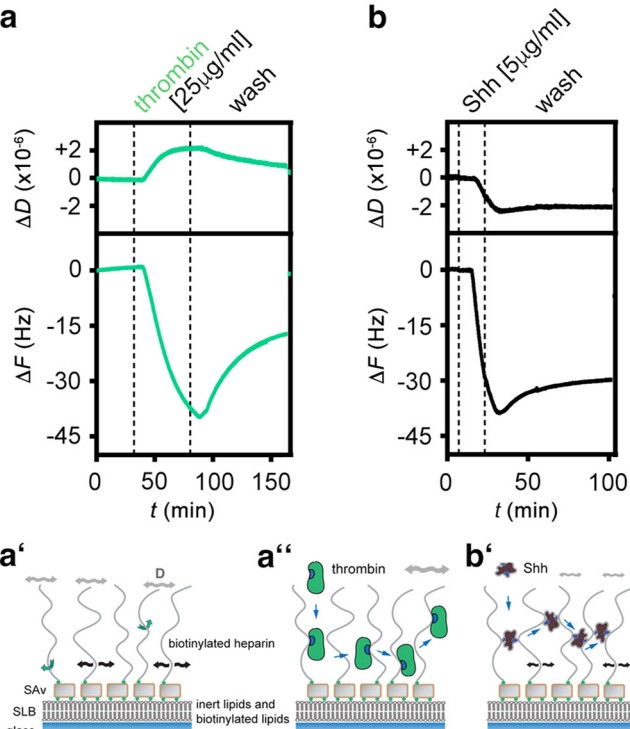

**Fig. 4 | Substitution of HS-binding Hh amino acids affects *Drosophila* wing patterning. a** Hh produced in clones of the posterior wing disc compartment (as shown in the cartoon, Hh is produced in the posterior compartment of the disc and moves over a significant distance[35] to the anterior receiving compartment, Supplementary Fig. 5) directly patterns the central domain of the adult wing (orange). Wing patterning beyond the central domain (the green area and the anterior top space that is not colored) depends on Dpp expression. Note that wing-mispatterning phenotypes are similar if due to the lack of Hh production in wing disc clones homozygous for hh[KO] or if homozygous for hh[KO;hh[R/E]]. Clonal hh expression, in contrast, completely restored wing patterning. **b** Quantification of wing phenotypes as shown in (**a**). Wings of female flies developing at 25 °C were analyzed, and Hh function during development was expressed as L3-L4/L2-L3 intervein area ratios. Error bars represent standard deviation. Statistical significance was determined by one-way ANOVA followed by Dunnett's multiple comparison test, $F = 152$. $n = 10$ (*hh*), $n = 26$ (hh[KO]), $n = 39$ (hh[KO;hh]), $n = 34$ (hh[KO;hh[R/A]]), and $n = 43$ (hh[KO;hh[R/E]]). ****$p \leq 0.0001$, **$p = 0.0019$, ns: $p = 0.96$. Source data are provided as a Source Data file. No sex bias was observed in any assay. Ctrl: FRT82B donor line.

**Fig. 5 | QCM-D analysis of Shh interactions with heparin. a–a″** Schematic representation and real-time analysis of thrombin binding to the functionalized QCM-D sensor surface. a′ represents the HS/heparin model matrix assembled on a silica surface. The supported lipid bilayer (SLB) exposes 5% biotinylated lipids that bind to streptavidin (SAv) linked to biotinylated heparin. Green arrows indicate free rotation of the coupled heparin chain, black arrows lateral movement on the SLB, and gray double-headed arrows indicate softness of the matrix (as sensed by increased dissipation, $\Delta D$). **a** Thrombin binding decreases $\Delta F$ during the protein incubation step and increases $\Delta D$ correspondingly, demonstrating protein binding and retention of a relatively soft heparin/thrombin layer (a″). **b, b′** Similar protein loading of the functionalized matrix ($-\Delta F$ about $-40$ Hz in both cases) by Shh is achieved more quickly (as indicated by faster $\Delta F$ decrease) and is associated with a negative $\Delta D$, indicating that Shh rigidifies the heparin layer. The film rigidification is due to heparin cross-linking (as confirmed by complementary FRAP assays, see Supplementary Fig. 6c). Representative graphs of three independent experiments are shown.

soluble heparin again served as an effective acceptor, rapidly eluting most Shh from the chip surface. In contrast, none of the selectively desulfated heparin variants eluted Shh from the chip surface, and chondroitin sulfate glycosaminoglycans also failed to increase Shh release from the chip surface (Fig. 7b). Taken together, this suggests a possible mode of directional protein transport in the extracellular matrix in vivo that may drive Hh molecules toward acceptors with increased overall charge compared with the donor HS, and away from HS donors with decreased overall sulfation.

## Discussion

Diffusion is one conceivable way to relay signaling molecules and morphogens from producing to receiving cells[8,9,11]. Yet, timely and reliable paracrine Hh morphogen function through extracellular 3D diffusion is difficult to envision if morphogen spreading were to occur out of the plane of folded epithelia such as in *Drosophila* wing and eye disks, because free diffusion in the entire extracellular space would dilute the morphogens and make them largely inaccessible to their cognate receptors. One previously suggested solution is that morphogens interact reversibly with non-signaling extracellular binders at the cell surface[11]. These interactions would largely restrict 3D Hh diffusion to 1D to 2D transport on the epithelial plane and thus allow most molecules to find their target Ptc faster[58–61]. Indeed, the essential prerequisite of restricted Hh transport, i.e., that Hh spread requires a continuous cellular layer or a continuous extracellular matrix, has recently been shown to be fulfilled[53]. However, the challenge remains to mechanistically explain how extracellular binders largely maintain diffusive Hh movement (instead of decreasing or halting it) to form dynamic, scalable, and robust Hh gradients.

We note that similar conceptional and mechanistic challenges have already been solved for intracellular proteins with the same efficiency problems. Since the 1970s, it has been recognized that the relatively short time that DNA polymerases, transcription factors, nucleases, and other DNA-binding proteins need to find their target cannot be explained by 3D diffusion and random collision: The association with a specific DNA target sequence is about two orders of magnitude higher[62] (summarized in ref.[63]). It was found that this striking increase is achieved by nonspecific, long-lived electrostatic interactions between the positively charged DNA-binding protein and the negatively charged DNA sugar-phosphate backbone[59,60]. The electrostatic attraction prevents protein diffusion away from DNA (or RNA[64]), yet does not keep the protein from moving along the axis of the double helix, effectively converting 3D into 1D diffusion, called sliding. An important observation was that sliding is further enhanced by fast and direct protein relays from one DNA sugar-phosphate backbone to the next to bypass obstacles. Such direct protein relay between DNAs requires two independent DNA-binding sites as a prerequisite for a process called "intersegmental protein transfer"[63]. During this process, one binding site remains associated with one DNA

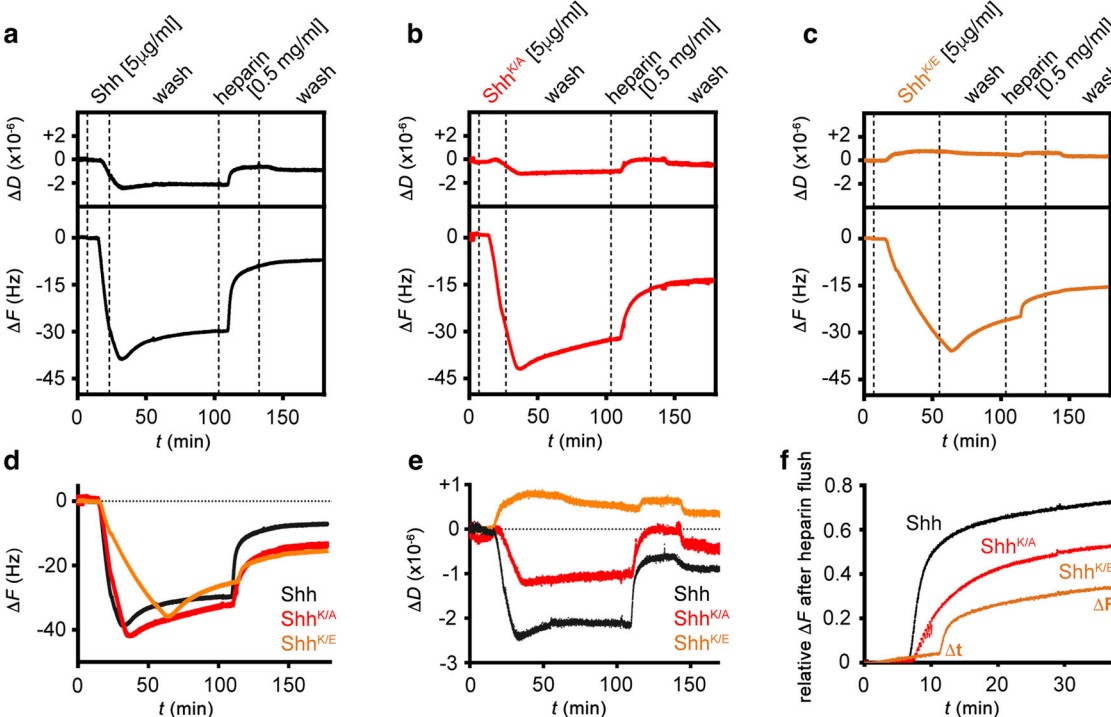

**Fig. 6 | The second HS-binding site in Shh facilitates cross-linking and switching between heparins. a–c** QCM-D binding assays analogous to Fig. 5, for Shh (**a**), Shh$^{K/A}$ (**b**), and Shh$^{K/E}$ (**c**), with an added step of competition with soluble heparin. **d** Overlays of frequency responses as shown in (**a–c**) demonstrate a reduced binding rate and increased unbinding rate in the wash buffer of Shh$^{K/E}$, and reduced unbinding of Shh$^{K/A}$ and Shh$^{K/E}$ upon competition with soluble heparin, compared to Shh. **e** Overlay of associated dissipation shifts as shown in (**a–c**) evidences reduced rigidification of the heparin layer by Shh$^{K/A}$, and even more so by Shh$^{K/E}$, compared to Shh, demonstrating that the second HS-binding site is essential for HS cross-linking. **f** Overlay of frequency responses (relative to full protein elution) from the start of soluble heparin-mediated protein elution ($t = 0$) until $t = 40$ min after elution start evidences an overall reduced elution rate for Shh$^{K/A}$ and Shh$^{K/E}$ compared to Shh, thus demonstrating that the second HS-binding site of Shh promotes rapid HS switching.

strand while the other binding site conducts target search as a pre-requisite for protein brachiation and intersegmental transfer[65].

From these established mechanisms, we suggest that direct electrostatic interactions with sugar-sulfate HS chains at the cell surface initiate and guide extracellular Hh transport in a similar manner to constrict Hh diffusion, possibly to 1D on HS chains and importantly to 2D on the HS-rich epithelial surface (Fig. 7c). In addition to the possibility of both Hh binding sites interacting with one (the same) HS chain, our work suggests that Hhs may also directly bridge the gaps between the many HS chains in the gradient field—severe obstacles if 1D or 2D transport is required for efficient protein spread while the 1D/2D axes are fragmented—via two sites of densely arranged positive charge to allow for intersegmental Hh transfer. We support this mechanism by showing that the reduced charge in the second HS-binding site strongly impairs Shh transfer between HS chains in vitro and interferes with several HS-linked Hh signaling-dependent developmental processes in vivo (Fig. 7d). Our findings thus provide four main mechanistic insights into how extracellular HS-non-signaling interactors may affect morphogen gradient formation.

The most important insight is that the propensity of Hhs to switch between HS chains, and the time required for intersegmental Hh transfer, represents a previously unknown essential determinant of temporally encoded morphogen movement and signaling[52]. This especially would apply to temporal gradients emanating from a static source and having to reach target cells at significant distances within a preset time frame. Unimpaired formation of such extended gradients requires multiple fast Hh switches between HS chains, e.g., into the anterior compartment of the developing wing disc: Over a distance of 50 μm, with HS lengths of ~100 nm per chain (an average number, HS lengths vary between 40 nm and 160 nm or sometimes more[66]), this

translates into 500 required Hh transfers at the least. During ocellar development (over 40 μm), this would translate into >400 required transfers, and multiple rounds of the self-reinforcing Wg/Hh signaling loop in the *Drosophila* embryo, while set in space, also requires numerous repeated rounds of Hh secretion and spread to stabilize the parasegment boundary. Note that delayed Hh switching propensity and time, as measured by QCM-D in our work, was determined for only one switching event. Thus, reduced switching propensity and speed will most severely affect the patterning of the wing disc or parts of the eye disc that depend on serial switching, or the formation and stabilization of embryo parasegment boundaries by repeated rounds of the self-reinforcing Wg/Hh loop. In contrast, cell determination by a one-time and direct cell-to-cell signaling event, such as during morphogenetic furrow progression, depends much less on Hh switch speed because of the shorter distances covered. This leads to only moderately impaired ommatidia numbers in the compound eye compared with more significant ocelli loss or L3-L4 intervein wing tissue loss, as shown in this study.

The second important aspect of our work, which is directly linked to the first, is that it resolves the apparent paradox that Hh binding to HS-non-signaling receptors at the cell surface seemingly facilitates Hh transport instead of slowing it down. The latter scenario applies to several soluble proteins and their interactions with non-signaling HS receptors, e.g., receptors with some $k_{on}$, some $k_{off}$, and no $k_{int}$ for internalization (note that the HS-bearing Glps have no transmembrane and cytoplasmic domains, and therefore are not specifically endocytosed)[10,67,68]. Here, the HS association of the ligand and gradient range are conflicting constraints because increasing the avidity of on/off binders to HS will shorten the gradient and vice versa[11]. Our explanation is that HS immobilization of soluble on/off binders

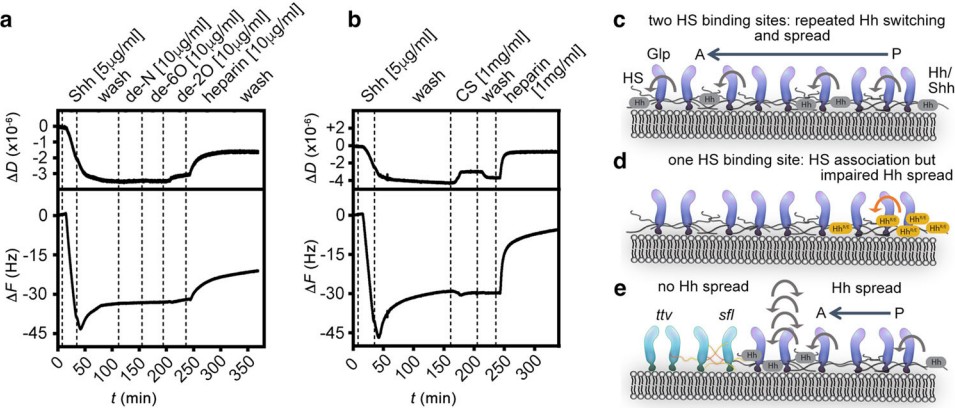

**Fig. 7 | Shh fails to cross-link and switch from heparin to lower sulfated soluble variants. a** In stark contrast to soluble heparin added to the wash buffer A, Shh does not transfer to selectively desulfated heparins with a less negative net charge. **b** Shh does not switch to low-sulfated chondroitin sulfate (CS, a related type of glycosaminoglycan consisting of repeating sulfated galactosamine-glucuronic/iduronic disaccharides instead of sulfated glucosamine-glucuronic/iduronic disaccharides for HS), even at high CS concentrations. **c** Model of direct repeated Hh switching. Two binding sites for HS facilitate Shh diffusion in an HS matrix through competition and direct repeated switching between neighboring HS chains (bent arrows). **d** Functional impairment of one binding site reduces the propensity for direct competition, and thus the rate of switching between HS chains, apparently slowing Hh/Shh spread down. This is expected to result in a shorter and steeper gradient. **e** High-affinity HS association restricts Hh/Shh movement to the cellular surfaces where suitably sulfated HS resides (the "diffusible zone"). Hh/Shh movement is effectively confined within the zone in two dimensions, with a gradient forming from posterior producing cells (right) to anterior receiving cells (left). Undersulfated HS are poor acceptors for Shh if already bound to higher sulfated HS, which explains that clones deficient in HSPG expression block Hh transport from HS-expressing cells in the *Drosophila* wing disc[15]. In the absence of properly sulfated (*sfl*) anterior HS, or without HS (*ttv*), Hh clusters remain confined to cell surfaces with appropriately sulfated HS (reversed arrows)—representing the constricted diffusible zone. Fully functional GPI-linked Glp-HS chains at the plasma membrane are shown in dark blue; non-functional variants lacking HS chains or linked to undersulfated HS in light blue. P posterior, A anterior.

effectively restricts their movement, which primarily occurs in an unbound (off) protein state via 3D diffusion. Hhs, in contrast, remain permanently bound to HS (on-state) while still being able to swiftly switch between the individual chains by using the second HS-binding site without having to go off-state. This represents an altogether different transport mode that eliminates the tradeoff problems as described for on/off binders. Note that our proposed mechanism still allows Hhs to interact with Ptc receptors during the time that the second HS-binding site, which also serves as a Ptc receptor-binding site, is unoccupied during transport. Hhs can thus choose between HS or Ptc, both without having to detach from the first HS. Note also that because of their high mobility in the plasma membrane (diffusion constant ≈1.5 μm² s⁻¹), GPI-anchored Glps may contribute to Hh transport on the surface of single cells[11], yet not between cells.

The third important insight from our work is that temporally encoded morphogen switching is in principle scalable. It was previously shown that Hhs do not relay through clones deficient in HS biosynthetic genes in the *Drosophila* wing disc, such as *sfl* and *ttv* (Fig. 7e)[15]. Our in vitro QCM-D findings using Shh and selectively desulfated heparin as the soluble phase (Fig. 7a) explain these observations, showing that Hh molecules are unable to switch from highly negatively charged HS to lower charged acceptor forms (e.g. N-desulfated *sfl*-like heparin), or to fields that altogether lack HS (e.g., *Drosophila ttv* wing disc clones), as simulated by the buffer washes during QCM-D. Notably, enhanced or reduced Hh signaling on a more graded scale was observed when HS sulfation in the *Drosophila* wing disc was more moderately changed[69]. In this previously published study, the removal of *Drosophila* HS 6-O desulfatase (*DSulf*, a secreted enzyme that removes sulfates from extracellular HS) in the posterior wing disc compartment was shown to increase HS 6-O sulfation and overall negative charge in Hh-producing cells. Interestingly, this restricted the gradient and lead to Hh loss-of-function wing phenotypes (comparable to the phenotypes that we show in Fig. 4). We explain this observation by impaired switching of endogenous Hh from HS in the producing field into the less negatively charged receiving compartment (Fig. 7a, e). Consistent with this possibility, *DSulf* removal and increased HS sulfation in the Hh receiving anterior compartment expands the gradient and leads to Hh gain-of-function wing phenotypes. The opposite phenotypes are observed upon *DSulf* overexpression: anterior *DSulf* overexpression impaired endogenous Hh function (again, HS in the receiving compartment becomes less sulfated and inhibits Hh spread), and posterior *DSulf* overexpression strongly enhanced endogenous Hh function (here, HS in the receiving compartment becomes more strongly sulfated and facilitates spread). Together with the fact that *DSulf* is dynamically expressed during wing disc development[69], our findings suggest that relative charge differences in the gradient field can regulate the propensity of Hh to switch, as well as the direction to which Hhs may preferably switch, without intermittent protein diffusion that may lead to its loss from its "diffusible zone". Both processes, in turn, may help scale the Hh gradient.

Finally, it has been noted that all known secreted morphogens interact with non-signaling binders in some way and often with HS[11]. Another important example of HS-binding molecules are chemokines that form HS-dependent haptotactic gradients for leukocyte recruitment. Mutagenesis of the HS-binding sites of three chemokines, monocyte chemoattractant protein-1/CC chemokine ligand (CCL)2, macrophage-inflammatory protein-1β/CCL4, and RANTES/CCL5, does not impair their chemotactic activity in vitro, but strongly affects cell recruitment when administered intraperitoneally[36]. Notably, monomeric variants of these chemokines, although fully active in vitro, are inactive in vivo[36]. To us, this suggests that protein homomultimerization may represent another possible mode to convert monomeric on/off binders into dimers or multimers with more than one HS-binding site that can only then form functional gradients. This leads to the general concept that charged HS may predetermine "diffusible zones" in tissues that guide proteins with patches of opposite charge on more than one site of the molecule (or dimers/multimers), while accumulating proteins with only one prototypical binding site such as growth factors. In addition, HS "microspaces" of increased HS sulfation over their surrounding tissue may accumulate chemokines, morphogens, or growth factors, independent of their transport by brachiation or on/off binding, to define "diffusible zones" for these proteins (that are restricted by neighboring tissues expressing HS with lower sulfation, or charge, possibly even in 3D). This may provide an

evolutionary strategy to steer different diffusible proteins differently to establish specifically regulated spatiotemporal signaling in multiple contexts within the same organism. It also calls for further efforts in the understanding of regulated HS biosynthesis and extracellular modulation of its sulfation as fascinating determinants of development and adult physiology.

## Methods

### Homology modeling, refinement, and evaluation

Homology models of Hh protein were constructed with Modeler 9.13 software[70] and the SWISS-MODEL Server[71]. A multiple sequence alignment with different template proteins (PDB entries: 2ibg[72], 3m1n[73], 1vhh[74], 2wfq[75], and 3k7g, accession codes, respectively) was used as input for the Modeler program and 10 comparative models were generated. The *loopmodel* class from Modeler was used for the subsequent refinement of C-terminal residues (residues 245–250, *Drosophila* nomenclature). Further, the best model (called model M in this work) was selected on the basis of Modeler scores and the Molecular Objective Function (Supplementary Fig. 1). The template used for the SWISS-MODEL Server was the 3m1n structure and only one model was generated (called model S). For evaluating the correctness of the built models (model M and model S), the programs PROCHECK and ERRAT[76] were used. Ramachandran plots obtained by PROCHECK showed that in both model M and model S, more than 90% of the residues were found in the most favorable and allowed regions. ERRAT provided overall quality factors of 95.48 for model M and 97.35 for model S. The evidence suggests that the two selected protein models (model M and model S) were acceptable and of good quality. The crystal structure of human Shh (PDB: 3m1l)[73] and modeled Hh were displayed by using the PyMOL Molecular Graphics System, Version 1.3, Schrödinger, LLC.

### MD simulations

The MD of ShhN (3m1n, monomeric) and the two modeled Hh structures (model M and model S) were simulated with the GROMACS 4.6.5 package. Each system was simulated in duplicate with the following protocol. Initial protein structures were solvated in a rhombic dodecahedron box of SPC/E water with a minimum of 1.0 nm distance between the protein and the faces of the box. Residues were assumed to be protonated according to their normal states at pH 7, with the exception of histidines in ShhN. The protons were assigned to histidines after inspection of H bond patterns in the ShhN 3m1n structure to the following nitrogens: $N_{\epsilon 2}$ for H134, H181, and H183; $N_{\delta 1}$ for H141; and both for H135. $Na^+$ and $Cl^-$ ions were added to neutralize the system at an ionic strength of 0.15 mol/L. The Particle Mesh Ewald method was used to compute electrostatic interactions under periodic boundary conditions. Structures were energy minimized and equilibrated by MD for 1 ns. Production simulations were run on a GPU for 200 ns with a time step of 2 fs. NPT conditions were stabilized at 300 K by a V-rescale thermostat, and at 1 atm by a Parrinello-Rahman barostat. Bonds were constrained by using the LINCS algorithm. The first 10 ns of all trajectories were dismissed to avoid any noise from the equilibration process. Only the last 190 ns were considered for the analysis, giving a total of 380 ns per system. Representative structures were extracted from trajectories with the g-cluster tool provided by GROMACS, based on mutual root mean square deviations. The structure shown in Fig. 1c, d is close to the cluster center and represents about 90% of the trajectories. The distance between the center of mass of the two HS-binding sites was measured with the GROMACS tool g_dist. R version 3.01 was used for all statistical analyses.

### Electrostatic calculations

Electrostatic potentials were computed by solving the non-linear Poisson-Boltzmann equation with APBS[77], using a grid with 0.08 nm spacing, relative dielectric constants of 79 and two outside and inside of proteins, respectively, and a water probe of radius 0.14 nm. The temperature was set to 298.0 K and ionic strength to 0.15 mol/L NaCl outside proteins and 0 mol/L inside proteins. Charges and radii were assigned to the structure with PDB2PQR v.1.9[78] using AMBER99 force field parameters, with manually edited van der Waals radii of 0.124 nm for the zinc ion and 0.02 nm for hydroxyl hydrogen atoms on Ser/Thr/Tyr amino acids. Energy grids were computed by Epitopsy commit revision d283db1[38] using the electrostatic potential grids, two heparin disaccharide conformers from[38] as probes, and the Fibonacci Spin rotation protocol with 150 unit vectors and 4 spins per unit vector. A model of human ShhΔ191–197 lacking the unstructured C-terminal peptide was extracted from chain B in the pdb entry 3m1n[73], while model S and model M were used for *Drosophila* HhNΔ251–257; alanine and glutamate mutants were created using the mutagenesis tool in the PyMOL Molecular Graphics System, Version 2.5.0, Schrödinger, LLC: R238A, R239A for Hh[R/A], R238E, R239E for Hh[R/E], R93A, R95A, R97A for Hh[CW/A], R93A, R95A, R97A, R238A, R239A for Hh[R/A;CW/A], K178A for Shh[K/A], and K178E for Shh[K/E]. The $\Delta\Delta G^{elec}$ values were calculated as the difference of $\Delta G^{elec}$ (obtained by a LogSumExp of all energy grid points) between each mutant protein and its corresponding wild-type protein.

### Cloning of recombinant Hh

We used murine Shh[79] and Hh sequences (nucleotides 1–1416, corresponding to amino acids 1–471 of *D. melanogaster* Hh) and HhN sequences (nucleotides 1–771, corresponding to amino acids 1–257) that were generated from *D. melanogaster* cDNA by PCR, using primer sequences that can be provided upon request. PCR products were inserted into pENTER for sequence confirmation and site-directed mutagenesis and subsequently into pUAST for protein expression in S2 cells or via Gibson cloning into pRIV[white34] for the generation of flies expressing the transgene under endogenous hh promotor control (Supplementary Fig. 3). Mutations in the second HS-binding site were introduced via the QuickChange Lightning site-directed mutagenesis kit (Stratagene, La Jolla, USA). Clonal analyses in *Drosophila* eye and wing disks required the generation of HackGal4FLP vector from the Hack-Gal4 Vector pBPGUw-Hack-QF2 (Addgene #80276)[80]. Briefly, we deleted two undesired FRT sites of the vector (one FRT at position 2021–2054 by Bgl II-mediated restriction at nucleotides 2015 and 2073, followed by restriction and religation, and the other FRT site at position 7323–7356 by recloning both flanking fragments, followed by QuickChange mutagenesis to remove two undesired AatII restriction sites at positions 1895 and 5898, and finally the replacement of the QF2-ORF by directed cloning of an AvrII-FLP-ORF-AatII fragment from *pCaSpeR*, yielding the construct HackGal4FLP.

### Cloning of Shh variants and expression in *Escherichia coli*

After we cloned ShhN wt (Δ191–198) into pGEX4T1, QuickChange mutagenesis was used to substitute a thrombin cleavage site of the vector with a TEV protease cleavage site (peptide sequence: ENLYFQS). This template was then used to generate Shh mutant variant sequences by QuickChange mutagenesis. *E. coli* BL21 were transformed by using heat shock transformation and plated out on ampicillin LB agar plates. Overnight cultures were grown in 50 mL TB medium in a shaking incubator at 37 °C and 150 rpm. The next day, 400 mL TB medium in 1000 mL shake flasks was inoculated with 15 mL of the overnight culture and again incubated at 37 °C and 150 rpm. After an $OD_{600}$ of ~1.2 was reached, expression of Shh or Shh variant proteins was induced by isopropyl-β-D-thiogalactopyranosid (IPTG) to a final concentration of 400 mM. Two hours later, the same amount of IPTG was added and the incubation continued for another 4 h. Cells were harvested in 50 mL falcon tubes and centrifuged at 4100 × g and 6 °C for 20 min. The supernatant was discarded and the step was repeated four more times. Cell pellets were stored at −20 °C until further use.

## Purification of ShhN

Reagents and solutions were kept on ice at all times. Cell pellets were carefully thawed on ice and cells were resuspended in 7 mL PBS supplemented with 4 μL NP-40 (Thermo Scientific) and 100 μL cOmplete protease inhibitor cocktail (Sigma-Aldrich), 1 μL DNase I (10 mg/mL), lysozyme (Roth), and 800 μL glycerol (Roth). Cells were lysed by sonication (60% duty cycle, 70% amplitude) for 3 × 1 min with cooling steps on the ice for 1 min between each step. Lysed cells were centrifuged for 20 min at 4100×$g$ and at 6 °C. The supernatant was first sterile filtered through a 0.45 μm filter and subsequently through a 0.2 μm filter. The filtered solution (5 mL) was then applied to an Äkta System (GE Healthcare) by using 1 mL GSTrap High-Performance columns (Cytiva) and a flow rate of 1 mL/min. The columns were washed with PBS for 30 min (1 mL/min buffer flow). The fusion protein was eluted with 10 mM glutathione in PBS for 15 min at a flow rate of 1 ml per min and fractionated. Fractions 9 and 10 were pooled and incubated with 10 units of ProTEV Plus (Promega) overnight on a shaker at 30 °C and 400 rpm to remove the Shh from the GST. The next day, Shh was dialyzed against cold MilliQ water by using a Slide-A-Lyser cassette (molecular weight cutoff 3.5 kDa, Thermo Fischer Scientific). After dialysis, samples were taken for SDS-PAGE analysis to determine protein purity and concentration by using BSA standards of known concentration followed by densitometric analysis. The remaining samples were aliquoted, lyophilized, and stored at −80 °C until further use.

## Expression and characterization of recombinant Hh in eukaryotic cells

S2 cells were cultured in Schneider's medium (Invitrogen, Carlsbad, USA) supplemented with 10% fetal calf serum. The cells were transfected with constructs encoding Hh and Hh variants, together with a vector encoding an actin-Gal4 driver using Effectene (Qiagen, Hilden, Germany), and cultured for 36 h in Schneider's medium before protein was harvested from the supernatant. Proteins used for protein stability assays were expressed, and supernatants containing wild-type and mutant proteins were collected, centrifuged at 13,000 $g$ for 20 min to remove cellular debris, and incubated at 37 °C for 0, 3, 6, and 12 h before precipitation with trichloroacetic acid and analysis by SDS-PAGE, Western blotting, and ImageJ (Version 2.9.0/1.53t) quantification as described below.

## Protein purification and analysis

Proteins were resolved by reducing 15% SDS-PAGE and immunoblotted onto PVDF membranes. The immobilized Hh proteins were detected with a primary polyclonal anti-HhN antiserum at 1:2000 dilution (rabbit IgG, Santa Cruz Biotechnology, USA) and visualized with a secondary peroxidase-conjugated donkey anti-rabbit IgG (1:5000) (Dianova, Hamburg, Germany) followed by chemiluminescent detection. The signals were quantified with ImageJ software.

## Synthesis of biotinylated heparin

Biotinylated heparin was synthesized by adapting a previously reported procedure[81]. A solution containing heparin (4 mM, Sigma-Aldrich) in 100 mM acetate buffer (made from glacial acetic acid (Carl Roth, Karlsruhe, Germany) and sodium acetate (Sigma-Aldrich) at pH 4.5) containing aniline (100 mM, Sigma-Aldrich) was prepared. Biotin-PEG$_3$-oxyamine (3.4 mM, Conju-Probe, San Diego, USA) was added to the heparin solution and allowed to react for 48 h at 37 °C. The final product was dialyzed against water for 48 h by using a dialysis membrane with a 3.5 kDa cutoff. The final solution was then lyophilized and stored at −20 °C. For further use, the conjugates were diluted to the desired concentrations in the buffer. The obtained biotinylated heparin was characterized by biotin-streptavidin binding assays using QCM-D. Low-sulfated HS was derived from porcine intestinal mucosa (Celsus Laboratories, Cincinnati, OH, USA), and high-sulfated HS was purified in the laboratory of Hughes Lortat-Jacob (Institut de Biologie

Structurale, Université Grenoble Alpes, Grenoble, France) and treated in the same manner. The average mass of all the heparin and HS isolates, when anchored to the surface, was estimated at 9 kDa (~18 disaccharide units) by QCM-D analysis[82].

## Preparation of small unilamellar vesicles (SUVs)

SUVs were prepared by adapting reported procedures[83,84]. A mixture of lipids composed of 1 mg/mL 1,2-dioleoyl-sn-glycero3-phosphocholine (DOPC, Avanti Polar Lipids) and 5 mol% of 1,2-dioleoyl-sn-glycero-3-phosphoethanolamine-N-(cap biotinyl) (DOPE-biotin, Avanti Polar Lipids) was prepared in chloroform in a glass vial. Subsequently, the solvent was evaporated with a low nitrogen stream while simultaneously turning the vial in order to obtain a homogenous lipidic film. The residual solvent was removed for 1 h under a vacuum. Subsequently, the dried film was rehydrated in ultrapure water to a final concentration of 1 mg/mL and vortexed to ensure the complete solubilization of the lipids. The lipids were sonicated for about 15 min until the opaque solution turned clear. The obtained SUVs were stored in the refrigerator and used within 2 weeks. For the FRAP experiments, lipid mixtures of DOPC and DOPE-biotin with DiD (Sigma-Aldrich) (molar ratio 94.9: 5.0: 0.1) were used to form the SUVs.

## QCM-D measurements

QCM-D measurements were performed with a QSense Analyser (Biolin Scientific, Gothenburg, Sweden) and SiO$_2$-coated sensors (QSX303, Biolin Scientific). The measurements were performed at 22 °C by using four parallel flow chambers and one peristaltic pump (Ismatec, Grevenbroich, Germany) with a flow rate of 75 μL per min. The normalized frequency shifts $\Delta F$, and the dissipation shifts $\Delta D$, were measured at six overtones ($i = 3, 5, 7, 9, 11, 13$). The fifth overtone ($i = 5$) was presented throughout; all other overtones gave qualitatively similar results. QCM-D sensors were first cleaned by immersion in a 2 wt% sodium dodecyl sulfate solution for 30 min and subsequently rinsed with ultrapure water. The sensors were then dried under a nitrogen stream and activated by 10 min treatment with a UV/ozone cleaner (Ossila, Sheffield, UK). For the formation of supported lipid bilayers (SLBs), after obtaining a stable baseline, freshly made SUVs were diluted to a concentration of 0.1 mg/mL in buffer solution (wash buffer A, 50 mM Tris, 100 mM NaCl (Sigma-Aldrich) at pH 7.4) containing 10 mM of CaCl$_2$ directly before use and flushed into the chambers. The quality of the SLBs was monitored in situ to ascertain whether high-quality SLBs were formed, corresponding to equilibrium values of $\Delta F = -24 \pm 1$ Hz and $\Delta D < 0.5 \times 10^{-6}$. Afterward, a solution of streptavidin (Sav; 150 nM) was passed over the SLBs, followed by the addition of biotinylated heparin (10 μg per mL). Each sample solution was flushed over the QCM-D sensor until the signals were equilibrated and subsequently rinsed with wash buffer A (see above). Before the addition of Shh protein (wildtype and mutants) solutions, the flow rate was reduced to 20 μL per min. For titrations, increasing concentrations of Shh were used as indicated, as well as for the heparin solutions. Analysis was performed using QSoft401 version 2.7.3.883. Selectively desulfated heparins (N-desulfated (DSH003/N), 6-O-desulfated (DSH002/6) and 2-O-desulfated (DSH001/2)) were biochemically characterized at and purchased from Iduron (Manchester, UK).

## FRAP measurements

For the FRAP experiments, SLBs were deposited in 18-well microslides with a glass bottom (Ibidi, Gräfelfing, Germany). Before the formation of the SLB, 150 μL of aqueous 2 M sodium hydroxide solution was added to the glass substrate for 1 h to form a hydrophilic surface. Afterward, the wells were rinsed three times with ultrapure water and three times with buffer (50 mM Tris, 100 mM NaCl, pH 7.4) containing 10 mM CaCl$_2$. Subsequently, a solution of 0.2 mg per mL SUVs in buffer with 10 mM CaCl$_2$ was added for 30 min at room temperature. An SLB was then formed by the rupture of SUVs onto the glass substrate.

Excess lipids were removed from the well by rinsing with 100 μL buffer five times. After the SLB formation, 50 μL buffer was left in the well plate in order to preserve the SLBs. Subsequently, a solution of Cy3-streptavidin (Cytiva, final concentration 1.5 μM) and biotinylated heparin (final concentration 100 μg per mL) was added to the well plates for 30 min. Finally, solutions of Shh (final concentrations were set to the twofold of concentrations given the half-maximal QCM-D responses (see Fig. 7a)) were added for 30 more min and the excess proteins was rinsed off five times with 100 μL buffer. With a confocal microscope, a circular spot of ~25 μm in diameter was bleached by using a laser at 552 and 638 nm (100% intensity), and then the fluorescence intensity in the bleached regions was monitored. For both the SLB and the SAv, the FRAP protocol consisted of 3 imaging loops (0.7 s intervals) before bleaching, 10 loops during bleaching (0.7 s intervals), and 20 loops during recovery (10 at 0.73 s intervals and 10 at 10 s intervals). All FRAP measurements were performed with a Leica SP8 confocal laser scanning microscope through a 63 × water objective. The Cy3 dye was excited with a 552 nm laser, and the emission was detected at 560–630 nm; the DiD dye was excited with a 638 nm laser, and the emission was detected at 650–700 nm. All images were analyzed by using Leica Application Suite X (LAS X) software version 3.7.1.21655. Data were plotted in Origin Lab version 2022b (9.9.5.167).

## Confocal microscopy of transfected cells

Transfected S2 cells were grown on gelatin-coated coverslips in 12-well plates. At 48 h after transfection, cells were washed with PBS, fixed with 4% PFA in PBS at room temperature for 10 min, and blocked with 1% BSA in PBS for 30 min. Cells were incubated with primary polyclonal anti-HhN antiserum (1:2000) (rabbit IgG, Santa Cruz Biotechnology, USA) at 4 °C for 12 h, washed three times with PBS, and incubated with secondary Cy3 donkey anti-rabbit IgG antibodies (1:600) (Dianova) for 2 h at room temperature. DAPI (1 μg per ml) was added to all incubations. Cells were washed three times with PBS and mounted in a mounting medium (Vector Laboratories). Images were taken on an LSM 700 Zeiss confocal microscope with ZEN software version 3.6. Orthogonal views of maximum intensity projections were created with ImageJ software.

## Confocal microscopy of *Drosophila* eye disks

Eye disks were dissected from wandering L3 larvae, fixed, permeabilized, and mounted. Samples were stained with anti-Eya antibodies (10H6 mouse, DSHB, 1:200, overnight), anti-Ci155 antibodies (2A1 rat, DSHB, 1:100, overnight), anti-GFP (rabbit, Invitrogen, overnight), and Alexa488-conjugated goat-α-rabbit (1:1000), Alexa568-conjugated goat-α-mouse (1:500), and Alexa647-conjugated goat-α-rat antibodies (1:300) (all Invitrogen) and DAPI. Disks were immunolabeled by using the same antibody batch and dilution, always following the same procedure. Images were taken on an LSM 700 Zeiss confocal microscope with ZEN software (Version 3.6), always with the same settings. Maximum intensity projections are shown.

## Protein bioactivity

Shh proteins were expressed in Bosc23 cells and secreted into the supernatants. Supernatants were added to C3H10T1/2 osteoblast precursor cells and their Hh-dependent differentiation was used as a readout to determine Hh bioactivity. Prior to analysis, protein aliquots were immunoblotted to confirm comparable protein amounts and protein integrity before use in the activity assay. The remaining conditioned media were then sterile filtered and applied to C3H10T1/2[85] cells in 15 mm plates. Cells were lysed 5 days after induction (20 mM HEPES, 150 mM NaCl, 0.5 % Triton X-100, pH 7.4) and the amount of alkaline phosphatase produced as a response of Shh-induced and Shh[K/A]-induced osteoblast differentiation was measured at 405 nm with 120 mM p-nitrophenolphosphate (Sigma) in 0.1 M glycine buffer at pH 8.5. All assays were performed in triplicate.

## Analysis of transgenic *Drosophila*

**Fly lines**. The following fly lines were used: hypomorphic hh[bar3] (FlyBase ID FBal0031487), hh null (hh[AC]/TM3, Bloomington #1749), nub-HackFLP (described below), nubbin AC-62 (nubbin-Gal4, Bloomington #25754), hh[KO;FRT]/TM3 (ref. [34], kindly provided by J.P Vincent), w[1118] and FRT82B kindly provided by C. Klämbt, FRT82B Rps3 (bearing Gal80, Gal80 not being used in this study as a tool) kindly provided by S. Luschnig. The nubHackFLP flies used also beared additional markers without influence on the results and of no relevance to the present study. Injection of pRIV[white] carrying the wild-type *hh* sequence or *hh* variants coding for R/A and R/E variant yielded hh[KO;hh]/TM3, hh[KO;hh[R/A]]/TM3, and hh[KO;hh[R/E]]/TM3 (this lab). For clonal analysis of homozygous alleles of all forms, a wing-blade-specific FLP source on chromosome II was generated. To this end, nubbin-Gal4 was chosen as the target for CRISPR-mediated gene conversion by using the pHack-Gal4FLP (described earlier) derivative of the Hack-Gal4 Vector pBPGUw-Hack-QF2 (Addgene #80276), following the method of ref. [80]. pHackGal4FLP was injected in a Cas9; nubbin AC-6-bearing background. Subsequent screening of the F1 offspring for positive convertants, as revealed by the presence of 3P3-DSRED, yielded 12 integration events from 500 injected embryos. 3P3-DSRED was then floxed out and isogenic lines were established. We refer to this chromosome as nubFLP. We used the following genotype for clonal analysis of the *hh* rescue constructs in the eye and wing disks (Fig. 3): yw eyFLP3.5/yw; nubFLP/+; FRT82B (Gal80) Rps3/FRT82B, hh[KO;hh[transgene] or no hh]. This enabled us to study clones in eyes and wings in the same animals. The resulting fly eye phenotypes were captured with a Nikon SMZ25 microscope and quantified by using Nikon F-package software version 4.5.01. w[1118] flies or heterozygous flies of the indicated genotypes served as positive controls. Wings of the adult F1 were collected and mounted in Hoyer's medium, and total intervein areas and total wing areas were quantified by MoticImage software version 2.0. All fly crosses and maintenance were conducted at 25 °C. For wing quantifications, ten male and ten female wings were analyzed for each data set and ratios between L3-L4 intervein areas and L2-L3 intervein areas were determined.

## Preparation of *Drosophila* embryos

Transgenic flies (hh[KO;hh[transgene]]/Tm3-GFP males and hh[KO; hh[transgene]]/Tm6-GFP females) were transferred onto apple juice plates covered with yeast in the center and placed in a 25 °C incubator for 24 h. After another 24 h incubation at 25 °C, all hh[KO;hh[transgene]] heterozygous and Tm3-GFP/Tm6-GFP larvae hatched and moved to the center of the plate. Homozygous embryos at the plate borders were collected and their genotypes were confirmed by the lack of GFP expression. They were dechlorinated in 50% bleach for 2 min while applying mild shear forces by pipetting the solution several times, and then fixed in a mixture of 400 μL PBS, 100 μL 37% formaldehyde, and 500 μL *n*-heptane. Embryos were washed in tap water and fixed again for 15 min on a shaker. Fixation was stopped by the addition of 1 mL methanol to the solution, vortexed to remove the extraembryonic membrane, and allowed to sit for 2 min to allow embryos to settle at the bottom of the tube. After removal of the liquid, embryos were washed three times with methanol, rehydrated in PBS, transferred into Hoyer's mounting medium, and analyzed with an Olympus BX60 microscope equipped with a dark field condenser lens and MoticImage version 2.0.

## Bioanalytical and statistical analysis

Most statistical analysis was performed in GraphPad Prism version 950. Applied statistical tests, post hoc tests, and the number of independently performed experiments are stated in the figure legends. A *P* value of <0.05 was considered statistically significant. Error bars represent the s.d. of the mean, unless noted otherwise. Differences in

protein decay between Hh and Hh$^{R/A}$ were analyzed using R package version 2.17.4 (http//:mc-stan.org/) and fitted with R package rstanarm, version 2.21.3. Hill fit for $K_D$ calculation was performed with the software OriginLab. The endpoint was set to 40 and the number of binding sites was set to 2.

## Reporting Summary

Further information on research design is available in the Nature Research Reporting Summary linked to this article.

## Data availability

Source data are provided with this paper.

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

## Acknowledgements
We are grateful to C. Klämbt and his group for their continuous support. The excellent technical and organizational assistance of S. Kupich and R. Schulz is gratefully acknowledged. We thank the laboratory of H. Lortat-Jacob (Institut de Biologie Structurale, Université Grenoble Alpes, Grenoble, France) for providing the high-sulfated HS purification, and R. Vivès (Institut de Biologie Structurale, Université Grenoble Alpes, Grenoble, France) for the disaccharide analysis of high-sulfated and low-sulfated HS. This work was financed by DFG (German Research Council) GRK1549/1 (K.G.), GR1748/7-1 (K.G.), CiM FF-2015-02 (K.G.), GR1748/8-1 (K.G.), GR1748/9-1 (K.G.), MA8629/1-3 (D.M.), UK Biotechnology and Biological Sciences Research Council (BBSRC) Research Grant—BB/X007278/1 (R.P.R.), UK Royal Society International Exchanges Award—IEC\R2\202035 (R.P.R.), and Mizutani Foundation for Glycoscience grant support (120010, K.G.).

## Author contributions
D.M., D.H., K.G., S.W., G.S., and R.P.R. designed the studies, F.G., J.F., D.D.I., D.M., G.S., J.-N.G., M.K., and K.G. performed the experiments, D.M., G.S., D.D.I., D.H., M.K., R.P.R., and K.G. carried out the analysis and K.G., D.M., R.P.R, and G.S. drafted and revised the article.

## Funding

## Competing interests
The authors declare no competing interests.
