## [Peer Review File · Nature Communications]

REVIEWER COMMENTS

Reviewer #1 (Remarks to the Author):

The work deals with the investigation of the role of Hedgehog (Hh) in differential specification of cell. The interaction among Hh and heparan sulfate (HS) were studied to understand how Hh binding to cell-surface-associated HS expands the Hh gradient. To this aim, QCM-D was applied as a sensitive tool to monitor real time the interaction among a QCM functionalized sensor (exposing heparins) with a circulating solution loading Shh protein (wild type and mutants) (or thrombin as control). Furthermore, the effect of circulating heparin on Shh binding was measured.

The use of the QCM-D technique is adequate to achieve the reported results and it allows to confirm the hypothesis with the unique ability to give a real time monitoring of the process occurring on the surface of the functionalized sensor.

The rationale of the QCM-D based experiments is exhaustively provided and results discussions are coherent with data obtained from QCM-D analysis.

Few additional information should be included to the work:

-The fifth overtone was used, authors should briefly justify this choice.

-The QCM sensor was functionalized forming a multilayered surface prior to analysis. The process of the multilayered surface formation can be followed by the QCM-D to monitor the multistructured surface formation. Indeed, this real-time monitoring of multilayers is one of the key elements of this technique. I suggest authors to include a brief discussion on this topic to include the evidence that the process was successful coating the sensor in a repeatable and homogeneous manner (added as SI).

- Authors used Shh instead of Drosophila Hh, however no evidences of these similarities are reported.

Reviewer #2 (Remarks to the Author):

In the manuscript titled "Direct and dynamic switching between heparan sulfates shapes the hedgehog gradient", Gude, Frose & Manikoswki et al propose a model for the biochemical basis of hedgehog diffusion through the extracellular matrix. The hedgehog field has been intensely focused on understanding whether hedgehog signaling occurs via extracellular diffusion, or whether the hedgehog protein is delivered by membrane projections. In their genetic and biochemical analysis, the authors convincingly advance the model that hedgehog diffusion is important for patterning by relating biochemical observations about hedgehog diffusion to genetic observations about Drosophila development.

It has been widely observed that secreted signaling proteins tend to have exposed positively charged amino acids. Here, the investigators noted that hedgehog and its conserved family of signaling proteins contains two positively charged patches, including a well-characterized Cardin Weintraub motif and a second patch ~30 nm away that has not been extensively described. Unlike the Cardin Weintraub motif, the novel positive patch (R238, R239) is not involved in binding the Patched receptor, meaning that its function can be investigated without compromising hedgehog signaling. In Drosophila, hedgehog is known to form patterns across a variety of lengthscales, ranging from ~1-2 cell diameters to >10 cell diameters. In the remainder of the manuscript, the authors describe how mutations that affect the charge on the positive patch are deficient in forming long-range gradients, but can nevertheless form short range gradients in natural fly tissues. Additionally, they relate these genetic observations to measurements of molecular diffusion and heparin interactions, and demonstrate that the mutants in which hedgehog cannot form

long-range signaling gradients have a corresponding decrease in their ability to crosslink heparin. Their findings suggest that bivalent interactions with heparin are required for long-range hedgehog activity, and support the model that hedgehog diffuses through the extracellular matrix by interacting transiently with heparin.

Overall, the manuscript considerably advances the field both materially and theoretically, and is technically well executed. The major strength of the manuscript is in the way it uses both genetic and biochemical tools to relate developmental phenotypes to molecular mechanisms. Theoretically, the manuscript advances the idea that transient multi-valent interactions with the extracellular diffusion, and this idea will likely be the basis of future investigations across a range of signaling protein families and tissue contexts.

In addition to the general comments above, I call out some minor concerns with the manuscript and data presentation below:

Figures

- Fig 1E – molecular dynamics-based calculations for the Hh R/E vs R/A alleles would be useful. Later figures establish that R/E mutants are more severe than R/A mutants. It would be useful to know if this difference reflects a difference in the K_d between Hh/Heparin disaccharide.
 - o In Fig S6A, the K_d for the WT Hh and for the R/E allele were measured experimentally. How do these experimental results compare to the predicted result based on MD in Fig 1E? Is data available for R/A, and are the results consistent between MD and direct measurement more generally?
 - o The methods section indicates the Hill coefficient was set to 2 for fitting Fig S6A. Is it possible to estimate the Hill coefficient from the data, rather than explicitly defining it based on the prediction?
- Fig 1F – data would be better plotted as a line graph, and the half-life would be better analyzed by measuring the slope of the [protein] vs log₂(time), using Fisher's R-Z to perform a statistical test.
- Fig 1G – this result is extremely important to the interpretation of the subsequent genetic data. Specifically, it would be extremely important to know if the K/E allele behaves similarly to the K/A allele. To quantitatively compare the bioactivity between wildtype and mutant Shh alleles, a single concentration is not sufficient if the concentration used is saturating the receptor, and it would be valuable to perform a full dose response curve for each ligand, to ask whether the EC₅₀ for each hedgehog allele is consistent. In addition, the left-most column is unlabeled.
- Fig 3 – the left-most column is also unlabeled in each plot.
- Fig 3 A,B – It is unclear to me from the schematics where the corresponding features are in the images on the right. For 3A, I can figure it out, but perhaps in 3B the schematic could better indicate what the images on the right capture in the schematic.
- Fig 6 E,F – it seems that the K/A retains considerable heparin-binding ability. If the molecular dynamics estimates of binding energy included both K/A and K/E alleles, it would be interesting to know if the binding energy ranks predicted by MD correspond to the ranks observed by QCM-D
- Figure S6C – why is there a corona of red signal in the FRAP experiments?

Text

- In line 117 and in the discussion, authors should soften the claim that Hh must interact with two heparin sulfate molecules in trans. They do not demonstrate that Hh interacts exclusively with two different heparin sulfate molecules, i.e. it is possible that Hh can interact with two different sites on the same heparin sulfate molecule.
- In general, it is very unfortunate that the biochemical experiments were performed using mammalian Shh instead of Drosophila Hh. If there is any more complete explanation for why the biochemical purification of Drosophila Hh failed, it would be useful in explaining why the project proceeded as described. Would it have been impossible or impractical to do the fly transgenics with mammalian Shh, to facilitate the direct comparison between biochemical and genetic data?
- The main text or Figure 5 would benefit from a concise description of how QCM-D measures binding and stiffness.
- Discussion could benefit from some speculation about how the biochemically complex

natural ECM might compare to the biochemically pure heparin used in these experiments. For example: if there were spectrum of heparin chain lengths or variation in the chain length across a tissue, would it influence patterning? Will any protein modified with heparin sulfate be a sufficient carrier for hedgehog? Etc.

- Discussion would benefit from further analysis of how variation in heparin sulfation across a tissue could contribute to morphogenesis by forming a "diffusible zone"

Overall, I found this paper to be an impressive interdisciplinary investigation of the basic biophysics of Hedgehog patterning. It effectively integrated evolutionary, genetic, developmental, biochemical and biophysical data and logic to convincingly demonstrate that heparin sulfate interactions are related to Hedgehog patterning, thereby supporting the model that the Hedgehog morphogen reaches its target cells by diffusion.

Reviewer #3 (Remarks to the Author):

1) General comments:

The paper "Direct and dynamic switching between heparan sulfate shapes the hedgehog gradient" by Gude et al. reports the direct interaction between Hedgehog (Hh) class molecules to negatively charged extra-cellular sugar chain, Heparan Sulfate (HS)/ Heparin requires specific amino acids in its N-terminal globular domain, in addition to the other known HS binding site named Cardin-Weintraub (CW) motif. By combining the molecular simulation, in vivo genetic models, and the relatively new developed technique to monitor the biophysical character of the molecules, the authors claim that the Hh binds two HS/heparin chains via two separately located HS-binding sites, and this "HS cross-linking" contributes to long-range Hh distribution. The authors propose Hh's switching between HS chains achieves long-range Hh distribution.

Although this study provides some vital knowledge about the biophysical insight of Hh interaction with HS class molecules, many of their conclusion is less supported by the experimental evidence. Thus, this reviewer thinks the current version of the manuscripts is too early to be published in Nature Communications and needs a substantial amount of additional data and modifications to the manuscripts.

2) Specific comments:

1. The authors assume that the binding of the mutated form of Hhs (Hh R/A, R/E, and Shh K/A, K/E) to HS was impaired because they lost the positively charged residues. Although this is a possible assumption, there could be several other possibilities. Is the structure of the protein not affected by this modification? Is the bioactivity of mutated Hhs identical to that of normal Hhs? For the latter possibilities, the authors provided some evidence. Still, these data are not enough, especially about the bioactivity of mutated *Drosophila* Hh (for bioactivity data, only the mutated Shh is provided). Indeed, their QCD-M analysis data suggest that the loss of binding ability is not simply caused by the loss of positive charge in the HS-binding site of Shh since the binding capacity of Shh K/A to HS seems not to be affected even though they decrease the positive charge in the binding site. For investigating the bioactivity of mutated *Drosophila* Hhs, it is adequate to perform S2 cell-based signaling assay by utilizing Hh signal indicators, such as phosphorylation or membrane localization of Smo protein and stabilization of Ci155. For cell-based signaling assay, Dejima et al. (JBC, 286, pp17103, 2011) provide an experimental system in which Hh extraction is not needed. Related to the above point, it is better to provide quantification imaging data about the extra-cellular Hh amount in Fig. S2.

2. The authors try to convince us that modification of HS affects Hh distribution, especially in long-range signaling. This reviewer thinks that their definition of "long-range" and "Short-range" is not clear since even in the case of morphogenetic furrow movement, Hh from posterior differentiating cells signals several cell diameters (as indicated by stabilization of Ci155, or expression of Hh target genes, *Ato*, shown in such as Baker et al. Dev. Biol., 335, pp356, 2009), and that is comparable the wing disc or ocellar specification cases. In addition to the distance between Hh providing cells to its receiving cells, the

amount of HS or expression of Glypicans may differ in each developmental context. Indeed, the authors showed mutated Hh fails to signal even in segment specification processes in embryos, that is short-range signal (Such as in Fig.2C). Their assumption for the effect of Hh mutation in the Hh distribution range would cause the reader confusion, thus needing a clear definition or explanation about the context in which Hh mutation affects its distribution.

3. Related to the above, experimental results with which authors evaluate how mutated Hhs affect its signaling range are based on the outcome of Hh distribution and not the distribution of the Hh signal range itself. Especially, the comparison between the number of ommatidia (range: 0- approx..700) and the number of ocelli (range: 0-3) is not fair to conclude that mutated Hh has effects only in long-range signal since the possible variance have a huge difference. This reviewer thinks evaluating how mutated Hhs affect its distribution and signaling range in imaginal discs is required. In addition, if the authors want to mention the difference between Hh K/A and K/E, it is appropriate to use Tukey's test, not Dunnett's test for multiple comparisons.

4. For QCM-D analysis in Fig.5, the authors conclude Shh "cross-link" HS-chains with the data in which adding Shh decreases dissipation (increasing stiffness). Is it a generally accepted assumption for QCM-D analysis? Is there no other possibility causing those results? For non-specialists, there needs some more explanation or evidence.

5. The authors claim that Shh cross-link two HS chains via its two independent HS-binding sites and "relay" between HS chains. They also claim that this "relay" contributes to obtaining the distribution "speed" of Hhs. This reviewer thinks this is an over-assumption that is not based on the experimental evidence. Indeed, their FRAP indicates that Shh with complete HS binding cite, which supposes to be "relayed" between HS chains, moves more slowly than the mutated Shh that is supposed not to be relayed. Authors need to provide experimental data and a clear explanation supporting their proposal.

Minor points:

1. Specific terms that first appear is needed to be defined. e.g., pdb in P5, L106.
2. It is not clear why three independent data for one genotype are shown only in Fig 2E. The authors need to explain this.
3. In Fig. 3 A and B, the labels of the first column in each panel are lacking.
4. For mosaic analysis of wing imaginal disc in Fig.4, authors need to mention or show data about the cell competition, since they mention it in the eye-imaginal disc case.
5. The preparation of de-60- and de-20- soluble heparin has not appeared in the material and methods. If the authors prepare those materials in their hands, the evaluation of prepared de-60 and de-20-soluble heparin is needed to be presented.

NCOMMS-22-25488A *Reply to reviewer's comments:*

We would like to thank reviewers for their time, effort and their constructive criticism. In our resubmitted manuscript, we have addressed all points raised by the reviewers:

Reviewer #1 (Remarks to the Author):

1) The fifth overtone was used, authors should briefly justify this choice.

ΔF and ΔD data was measured at six overtones ($n= 3, 5, 7, 9, 11, 13$) that showed qualitatively similar results. Because of its higher signal-to-noise ratio, we presented the fifth overtone ($n=5$) in our work. We included this information in the M&M section (line 654ff).

2) The QCM sensor was functionalized forming a multilayered surface prior to analysis. The process of the multilayered surface formation can be followed by the QCM-D to monitor the multistructured surface formation. Indeed, this **real-time monitoring of multilayers** is one of the key elements of this technique. I suggest authors to **include a brief discussion** on this topic to include the evidence that the process was successful coating the sensor in a repeatable and homogeneous manner (**added as SI**).

We now provide the requested quality control (routinely performed during functionalization of the chips) in revised Supplemental Fig. 6a and discuss the QCM-D functionalization protocol in the Supplemental Fig. 6a legend.

3) Authors used Shh instead of Drosophila Hh, however no evidences of these similarities are reported.

*We agree that it is very unfortunate that Drosophila Hh instability prevented us from using the purified protein variants for QCM-D analysis (Hh instability is probably due to the lack of the tetrahedrally coordinated zinc ion that stabilizes all vertebrate Hhs (PMID: **10500113**, **10555969**)). However, we note that the most important aspect – the essential role of HS expression for Hh biofunction in vivo – is conserved between vertebrates and invertebrates (Drosophila: PMID: **10549295**, Vertebrates: PMID: **15177029**). Moreover, it is known that Shh expression in the fly rescues early developmental defects in hh-deficient fly embryos (PMID: **8269519**), further supporting functional similarities between both proteins. From these observations, together with our in silico analyses, we conclude that vertebrate and invertebrate Hhs are functionally similar, and that Shh behavior in QCM-D can be expected to reflect that of the fly protein.*

Reviewer #2 (Remarks to the Author):

We thank this reviewer for highlighting the significance of our work and for the valuable suggestions. We addressed all suggestions and minor concerns in our revised manuscript.

General considerations on our modeling approach:

*Regarding the suggestion to expand our Hh-heparin binding analyses in silico, we have carried out in prior work (PMID: **29266929**) seven independent 0.5 microsecond all-atom molecular dynamics simulations of Shh lacking the CW domain (Shh Δ CW) with heparin disaccharide for a total combined time of 3.5 microseconds, which revealed several binding sites, one of which the heparin dimer was unable to escape from in 5 out of 7 simulations. This binding site is labeled cluster 3 in Figure S9B in Grad et al. 2018 (PMID: **29266929**), and corresponds to the Shh*

globular domain amino acids R154, R156 and K179 (human nomenclature, corresponds to R153, R155 and K178 in the mouse = second HS binding site in our work, Fig. 1). These amino acids are spatially close to the *Drosophila* Hh globular domain amino acids R238, R239.

These MD simulations took several weeks of computer time, and the 0.5 microsecond simulated time was too short to observe the unbinding event from the globular domain. Overcoming this undersampling issue would likely require MD simulations one order of magnitude longer, or accelerated molecular dynamics methods that rely on modified Hamiltonians. The computational cost associated with this type of simulation motivated us to develop a faster computational method, Epitopsy, that ignores the dynamics of the binding event, and instead scans a large number of binding modes using rigid bodies. We could show the probability of presence of the heparin ligand predicted by this method was in excellent agreement with the probability of presence obtained by concatenating all frames from the 3.5 microsecond MD simulations (Figure 2C-D and Figure 3B in Grad et al. 2018 (PMID: 29266929)). The small discrepancy was attributed to undersampling in the MD simulations.

The Epitopsy method calculates energies that can be used to estimate the electrostatic contribution of the $\Delta\Delta G$ for protein mutants. While these predicted $\Delta\Delta G$ values are not calibrated, and therefore not directly comparable to experimentally determined binding affinities, they are still useful to reveal trends in affinities between protein mutants.

We have decided to re-do the calculation for Fig. 1e using a newer version of Epitopsy, which has received several corrections since the original plot was made. In particular, the older version of Epitopsy did not correctly define the protein-solvent boundary, resulting in predicted binding energies with a high variance. The new plot in Fig. 1e reports more accurate predictions of the Coulomb contribution to the binding energy; these values tend to underestimate the binding energy, since they lack some contributions to the Gibbs free energy of binding (desolvation, re-orientation of the protein side-chains). The calculations also use *Drosophila* HhN model S and model M with point-mutations, instead of using a different ensemble of models for each mutant, so as to make the energies directly comparable between mutants. We also carried out the same calculations with human Shh K/A and K/E alleles, as suggested.

1) Fig 1E – molecular dynamics-based calculations for the Hh R/E vs R/A alleles would be useful. Later figures establish that R/E mutants are more severe than R/A mutants. It would be useful to know if this difference reflects a difference in the K_d between Hh/Heparin disaccharide.

We agree that the Hh R/E mutant should be part of this calculation. It has therefore been added to the plot (Fig. 1e, Supplemental Table 3), and the trend shows that Hh R/E should have a lower affinity to heparin disaccharide compared to Hh R/A. Therefore, our *in silico* calculations reflect that R/E mutants are more severely affected than R/A mutants *in vivo*, and are in line with the *in vitro* results obtained by QCM-D and FRAP. However, we need to point out this plot is not based on molecular dynamics simulation, but on protein-ligand electrostatic complementarity scanning calculations using Epitopsy, which has been shown to be consistent with long MD calculations of this system (Grad et al. 2018 (PMID: 29266929)).

2) In Fig S6A, the K_d for the WT Hh and for the R/E allele were measured experimentally. How do these experimental results compare to the predicted result based on MD in Fig 1E? Is data available for R/A, and are the results consistent between MD and direct measurement more generally?

In original Fig S6A, the reported K_d is actually for Shh, not Hh. However, as mentioned in our answer to question # 1 above, we have carried out the same electrostatics calculation using Shh and report in our revised manuscript (Fig. 1e) that the Shh R/E has a lower affinity to heparin disaccharide compared to Shh R/A. We can only compare trends, however, since the ΔG values determined *in silico* are not directly comparable with experimentally-determined ΔF values (including the newly added values of ShhK/A, now shown in Supplemental Fig. 7a, former Fig.

6a), as they represent a different quantity. In addition, the predicted ΔG values neglect some contributions to the overall affinity.

3) Comment: The methods section indicates the Hill coefficient was set to 2 for fitting Fig S6A. Is it possible to estimate the Hill coefficient from the data, rather than explicitly defining it based on the prediction?

We determined the Hill coefficient without previous presets from the data and found the determined value (2.4 ± 0.5) close to 2, which also resulted in a similar calculated Shh K_D (27 ± 2 nM).

General remark: The Hill coefficient only equals the number of binding sites if there is perfect positive cooperativity between the different binding sites and otherwise gives a lower limit of the number of binding sites (PMID: 9285481). The noise of the data and the measurement intervals do not allow determining the exact Hill coefficient but the assumption of setting it to two builds on the finding that Shh has at least two HS binding sites (PMID: 22049079, PMID: 24062467). Please note that our mutagenesis approach did not abolish the second HS binding but left it (partly) functional, as shown in Fig. 1 and the FRAP data in Supplemental Fig. 7. Still, we performed a Langmuir fit and found that the calculated changes in K_D remained within the margin of error (data can be provided upon request).

4) Fig 1F – data would be better plotted as a line graph, and the half-life would be better analyzed by measuring the slope of the [protein] vs $\log_2(\text{time})$, using Fisher's R-Z to perform a statistical test.

For the revision we analyzed the decay dynamics of Hh and HhR/A quantitatively with a Bayesian model (new Fig. 1s and Supplementary methods). The result of the analysis is consistent with decay dynamics that are indistinguishable between the two proteins.

5) Fig 1G – this result is extremely important to the interpretation of the subsequent genetic data. Specifically, it would be extremely important to know if the K/E allele behaves similarly to the K/A allele. To quantitatively compare the bioactivity between wildtype and mutant Shh alleles, a single concentration is not sufficient if the concentration used is saturating the receptor, and it would be valuable to perform a full dose response curve for each ligand, to ask whether the EC_{50} for each hedgehog allele is consistent.

We agree with this reviewer that unperturbed protein signaling capacities of mutant HhK/R-A and HhK/R-E proteins at receiving cells are essential for the interpretation of our in vivo results. For this reason, in the original manuscript, we have already demonstrated that all proteins restored eye development in flies transheterozygous for the Hh^{bar3} locus (Fig. 2d,e). In our opinion, this assay is the most valuable in supporting that arginines 238 and 239 are not essential for Ptc receptor binding and signaling, and that secretion/folding are not impaired by the mutations. Moreover, the activities of the HhR/A and HhR/E mutants in this assay shows that differences are minor ($n=600$ vs. $n=584$ ommatidia, $n=30$ ea, $p=0.03$). Still, we performed additional in vitro analyses, as requested by this reviewer and reviewer #3. These analyses are now shown in Fig. 1f-h.

For these assays, we first transfected eucaryotic Bosc23 cells, normalized Shh protein amounts secreted into the media by SDS-Page/Western blotting, and added different media amounts to C3H10T1/2 reporter cells. These assays confirmed similar biofunctions of all proteins and demonstrated that saturation of Ptc was not reached in our assays. Actually, we know from previous experiments that ShhN, a protein variant lacking its C-terminal cholesterol, is about 10x more bioactive than the dual-lipidated, authentic proteins if used in our C3H10T1/2 assay, and it is still not saturating the receptor. Therefore, we hope that the Shh dilution assay now shown in Fig. 1g is sufficiently convincing that Ptc saturation was not reached in our assays and that all protein variants show similar activities at the level of Ptc. In addition, we now also show that

invertebrate Hh is active in the same C3H10T1/2 assay, and that Hh variant activities are similar (Fig. 1h).

6) Fig 3 – the left-most column is also unlabeled in each plot.

We are sorry for this mistake and corrected it in revised figures 3 and 4.

7) Fig 3 A,B – It is unclear to me from the schematics where the corresponding features are in the images on the right. For 3A, I can figure it out, but perhaps in 3B the schematic could better indicate what the images on the right capture in the schematic.

We have now revised the cartoons in Fig. 3 to make the differences between both Hh-patterned fields clearer. We have also added confocal stacks of Drosophila eye discs expressing Hh or HhR/E to Supplementary Fig. 5 to better illustrate this developmental process. The reviewer may also find the attached movie helpful. This movie clearly shows the morphogenetic furrow but also the position of the prospective ocelli (white arrowheads in Supplementary Fig. 5).

8) Fig 6 E,F – it seems that the K/A retains considerable heparin-binding ability. If the molecular dynamics estimates of binding energy included both K/A and K/E alleles, it would be interesting to know if the binding energy ranks predicted by MD correspond to the ranks observed by QCM-D.

We re-calculated $\Delta\Delta G_{\text{elec}}$ for the different Shh variants and found that HS interactions of HhR/E are indeed more strongly impaired than those of the HhK/A variant (revised Fig. 1e). QCM-D calculations now shown in revised Supplementary Fig. 7 reflect this difference: we obtained 20.6 nM for the K/A variant, but 83.6 nM for the K/E variant. However, note that the kinetic data derived from the QCM-D frequency shifts should be considered 'effective' and interpreted with caution.

9) Figure S6C – why is there a corona of red signal in the FRAP experiments?

We noticed that different fluorophores used in our lab differ in this regard, but have not found any evidence that halo formation affects the experimental outcomes. Please also see data obtained using a different fluorophore in FRAP analyses now shown in Supplementary Fig. 6c.

Text

10) In line 117 and in the discussion, authors should soften the claim that Hh must interact with two heparin sulfate molecules in trans. They do not demonstrate that Hh interacts exclusively with two different heparin sulfate molecules, i.e. it is possible that Hh can interact with two different sites on the same heparin sulfate molecule.

We agree with this reviewer and softened the claim, now stating from line 122ff of the revised manuscript "This suggests that, in addition to the possibility that both sites cooperate in Drosophila Hh binding to only one HS chain, Hh may bind to two individual HS chains simultaneously." In the discussion (line 405ff), we address this question again. It should here be stressed, however, that our QCM-D data (showing Shh-induced HS film rigidification) and FRAP data (showing Shh-induced HS immobilization) jointly provide clear evidences for inter-chain cross-linking by Shh, at least temporarily.

11) In general, it is very unfortunate that the biochemical experiments were performed using mammalian Shh instead of Drosophila Hh. If there is any more complete explanation for why the biochemical purification of Drosophila Hh failed, it would be useful in explaining why the project proceeded as described. Would it have been impossible or impractical to do the fly transgenics with mammalian Shh, to facilitate the direct comparison between biochemical and genetic data?

We agree with this reviewer that the required switch from Hh used in vivo to Shh for our in vitro assays is very unfortunate. We tried for almost two years to obtain sufficient quantities of fly Hh for QCM-D and FRAP before switching to in the same expression system. We suspect that the lack of the otherwise highly conserved zinc ion in Drosophila Hh - that confers structural stability to the protein - is the most likely reason for our problems with expression and purification (PMID: 10555969). However, based on published findings that the role of HS in Hh biology is highly conserved, we believe that the fly protein would have behaved in a similar manner to Shh in QCM-D and FRAP.

12) The main text or Figure 5 would benefit from a concise description of how QCM-D measures binding and stiffness

We now explain the technology extensively in our revised manuscript (lines 300ff) and in the Supplementary dataset (Supplementary Fig. 6).

13) Discussion could benefit from some speculation about how the biochemically complex natural ECM might compare to the biochemically pure heparin used in these experiments. For example: if there were spectrum of heparin chain lengths or variation in the chain length across a tissue, would it influence patterning? Will any protein modified with heparin sulfate be a sufficient carrier for hedgehog? Etc.

It is well-established that not all HS-modified proteoglycans contribute to Hh signaling, and that glypicans Dally and Dally-like are most relevant for Hh morphogen transport and gradient formation (PMID: 14729575). The probable reason for this is that glypicans carry several HS-chains proximal to the cell surface, making them ideal candidates to keep Hhs close to the cell surface and the receptor Ptc. However, we would like to refrain from discussing this point in our revised manuscript, because it would lengthen the discussion considerably and may also distract from the importance of the HS chains and their sulfation degree in our model. Moreover, although increased HS chain length may well result in fewer switching events and may thus speed up transport, it is too much of a speculative idea at this point.

14) Discussion would benefit from further analysis of how variation in heparin sulfation across a tissue could contribute to morphogenesis by forming a “diffusible zone”

We agree that this is an important point that we indeed currently investigate in our lab. We therefore added this idea to the discussion, touching this issue but without stating too much on our currently investigated concepts (lines 480ff, 495ff and legend of Fig. 7).

Overall, I found this paper to be an impressive interdisciplinary investigation of the basic biophysics of Hedgehog patterning. It effectively integrated evolutionary, genetic, developmental, biochemical and biophysical data and logic to convincingly demonstrate that heparin sulfate interactions are related to Hedgehog patterning, thereby supporting the model that the Hedgehog morphogen reaches its target cells by diffusion.

We thank this reviewer for the very positive evaluation of our work, and for outlining its importance and potential.

Reviewer #3 (Remarks to the Author):

We thank this reviewer for the constructive criticism and agree that the careful functional characterization of all mutant Hh and Shh proteins is absolutely essential. In our revised study, we address these points in detail.

The authors assume that the binding of the mutated form of Hhs (Hh R/A, R/E, and Shh K/A, K/E) to HS was impaired because they lost the positively charged residues. Although this is a possible

assumption, there could be several other possibilities.” We addressed this point by careful (re)characterization of unimpaired Hh variant secretion to the cell surface (Supplementary Fig. 2 a,b), similar Hh variant expression levels and levels of protein solubilization (Supplementary Fig. 2c), similar protein stability in solution (Fig. 1f), similar Shh biofunction at the level of Ptc (Fig. 1g) and also similar Hh protein variant activities at the level of Ptc (Fig. 1h). Of note, these analyses now also contain the ShhK/E or HhR/E variants. We would also like to point out that protein variants, if expressed under endogenous promoter control from one allele in a hhbar3 background, restore Drosophila eye development (Fig2 d,e). In our opinion, this assay is the most convincing indication that protein processing, secretion, folding and Ptc binding/signaling are unaffected by the Hh R/A and R/E mutations, especially if taken together with the in vitro data newly added to the revised manuscript (Fig. 1g,h, Supplementary Fig 2).

Indeed, their QCD-M analysis data suggest that the loss of binding ability is not simply caused by the loss of positive charge in the HS-binding site of Shh since the binding capacity of Shh K/A to HS seems not to be affected even though they decrease the positive charge in the binding site”. We would like to reply that the capacity of ShhK/A to interact with HS in the QCM-D setup is affected, as clearly indicated by the detected increase in dissipation (Fig. 6e). However, we agree that HS binding of the R/A and K/A variants are less affected – which is in line with the much milder phenotypic changes observed for this variant in vivo (Fig. 3,4, Supplementary Fig. 4).

Finally, this reviewer suggests additional Hh activity measurements by using contact-dependent BMP signaling as a template to model Hh trans signaling in Drosophila S2 cells. However, we note that this technique requires the transfection of signal-receiving S2 cells with Mad-FLAG and of signal-sending S2 cells with HA-dpp, in addition to dally-myc HSPG (because, apparently, S2 cells do not express sufficient HSPG for juxtacrine signaling) in the original publication. To assess whether an adapted approach may be used to quantify Hh signaling, as suggested by the reviewer, we analyzed the expression levels of essential Hh pathway components in S2 cells by qPCR. The results are shown below and indicate that transfection of at least seven Hh pathway components would be required to reconstruct a (theoretically) functional signaling machinery in S2 cells – in addition to dally HSPG and hh co-expression. We do not believe that this is feasible, especially if taken together with the fact that the entire Dejima et al. paper is dedicated to the setup of the relatively easier BMP assay. We therefore refrained from trying to establish such an assay.

Instead, we expressed Hh and HhR/A and HhR/E variants in S2 cells and found that the invertebrate proteins effectively signalled to C3H10T1/2 reporter cells. We therefore now show in revised Fig. 1h that all Hh variants have similar activities at the level of Ptc.

Related to the above point, it is better to provide quantification imaging data about the extra-cellular Hh amount in Fig. S2.

We agree and now provide the requested data (including the HhR/E variant) in Supplementary Fig. 2b,c.

	S2 cell	Cq	Mean	DCq	
				Mean minus mean 18s	
	Hh	29,25	29,23	17,573	
	Hh (2)	29,69	29,43	17,107	
	lhog	36,87	36,92	25,263	
	lhog (2)	37,4	37,70	25,377	

	Boi	23,36	23,19	11,533	
	Boi (2)	23,08	23,00	10,677	
	Ptc	25,79	25,85	14,187	
	Ptc (2)	26,42	26,40	14,080	
	Smo	25,79	26,50	14,837	
	Smo (2)	26,09	26,19	13,873	
	Ci	35,64	33,94	22,277	
	Ci (2)	33,66	33,75	21,433	
	Fused	25,53	25,55	13,890	
	Fused (2)	25,78	25,87	13,547	
	18s	11,64	11,66		
	18s (2)	12,3	12,32		

qPCR of Hh pathway component expression in S2 cells. Results from two independent analyses are shown.

2) The authors try to convince us that modification of HS affects Hh distribution, especially in long-range signaling. This reviewer thinks that their definition of “long-range” and “Short-range” is not clear since even in the case of morphogenetic furrow movement, Hh from posterior differentiating cells signals several cell diameters (as indicated by stabilization of Ci155, or expression of Hh target genes, Ato, shown in such as Baker et al. Dev. Biol., 335, pp356, 2009), and that is comparable the wing disc or ocellar specification cases.

We agree with this reviewer that “long-range” and “short-range” have not been clearly defined. We therefore changed “long-range” and “short-range” into “up to 10 μ m (line 184), that require morphogen spread over relatively long distances of tens of micrometers (line 292)” and “tens of micrometers (line 1149)” including references to make it clearer that relative differences in transport range are investigated in our study.

4) In addition to the distance between Hh providing cells to its receiving cells, the amount of HS or expression of Glypicans may differ in each developmental context.

This is certainly true but will affect our different transgenes to the exact same extents, meaning that their spread can still be compared which leaves our conclusions unchanged.

5) Indeed, the authors showed mutated Hh fails to signal even in segment specification processes in embryos, that is short-range signal (Such as in Fig.2C).

Segment specification indeed requires Hh signaling over a short distance at the boundary, but in a self-reinforcing repeating manner. We therefore assume that the HhR/A or E / Wg signaling loop may slowly decrease in intensity over time.

6) Their assumption for the effect of Hh mutation in the Hh distribution range would cause the reader confusion, thus needing a clear definition or explanation about the context in which Hh mutation affects its distribution.

In our revised manuscript, we more clearly relate the specific developmental systems under investigation with the respective Hh variant effects.

7) Related to the above, experimental results with which authors evaluate how mutated Hhs affect its signaling range are based on the outcome of Hh distribution and not the distribution of the Hh signal range itself. Especially, the comparison between the number of ommatidia (range: 0- approx..700) and the number of ocelli (range: 0-3) is not fair to conclude that mutated Hh has effects only in long-range signal since the possible variance have a huge difference. This reviewer thinks evaluating how mutated Hhs affect its distribution and signaling range in imaginal discs is required.

We agree and now address this point in revised Supplementary Fig. 5. First, we provide evidence that the disc proper consists to the largest extend of clonal tissue derived from the hh allele bearing twin spots. The Minute twin spot is eliminated and has been cleared away by the time of our analysis. This is evident from the complete absence of ubiGFP in the disc proper for the entire part that gives rise to the eye and head vertex (Supplementary Fig. 5b). GFP positive heterozygous glia visible in the eye disc flooring is CNS derived and has undergone migration from the brain through the optic stalk to get in contact with the nascent photoreceptor. Other GFP positive cells are superficial on top of the imaginal disc (see movie S1), possibly Bolwig nerve and other tissue such as peripodial and developing muscle cells.

In Revised Supplementary Fig. 5b we now also show general hypotrophy of the eye and antennal part of hh KO disc, stalling of migrating glia en route and no eye absent (eya) domains indicating prospective ocelli development. The CI155 staining is present, yet it has no resemblance of the MF progression visible in discs expressing Hh or HhR/E. The antennal disc shows in part differential CI expression which makes it possible to judge that the staining per se is valid.

The RE eye discs show photoreceptor axon outgrowth and subsequent coverage by immigrated glia. In the area of prospective ocelli, however, there is loss of the pronounced eya domains and we also noted coinciding indents in the discs that are absent in Hh controls. The correlating CI155 stain is also altered notably in the prospective ocelli area. Incidences of slightly weaker eye disc phenotypes with incomplete loss of the eya domain make it conceivable why some ocelli may escape, but we cannot offer a straightforward explanation for that yet.

*General remarks to this reviewer: To date and to our knowledge, no systematic effort to recover hh deficient clones for end point analysis in a Minute background other than in hsFlp clones has been undertaken. hsFLP induced hh LOF Minute clones were used in wings as reference by Basler (PMID: **8145818**) showing a dramatic tissue loss phenotype. Burke et al (PMID: **10619433**) reported a moderate phenotype for a posterior hsFLP clone in the wing disc. By their nature hsFLP induced clones are variable as they largely depend on timepoint of induction with respect to animal staging, heat exposure method, onset and duration. In contrast, the strength of our approach is to yield consistent, reproducible and quantified results for all genotypes analysed (as now demonstrated in Supplementary Fig. 5).*

*It came to our attention that our HACK approach has been independently used elsewhere (PMID: **33444322**) to convert a Gal4 line to a Flp line, without further publication of details as of our resubmission date. The authors adopted the HACK approach further to generate CAS9 lines in a similar fashion (PMID: **33782117**). Clonal analysis, using edited alleles such as our RIV-edited hh variants has been performed before. Such modified alleles were first used in MARCM analysis for the ceramidase schlank (PMID: **29386138**).*

8) In addition, if the authors want to mention the difference between Hh K/A and K/E, it is appropriate to use Tukey's test, not Dunnett's test for multiple comparisons.

We did not compare every mean with every other mean, which would require a Tukey test. Instead, we compare every mean to a control mean, requiring the Dunnett test. Because Hh is used as the sole control mean in figures 2-4, therefore, we believe that using the Dunnett test is appropriate.

9) For QCM-D analysis in Fig.5, the authors conclude Shh “cross-link” HS-chains with the data in which adding Shh decreases dissipation (increasing stiffness). Is it a generally accepted assumption for QCM-D analysis? Is there no other possibility causing those results? For non-specialists, there needs some more explanation or evidence.

We added a brief description and relevant literature to our revised manuscript to clarify this point (lines 301ff and Supplementary Fig. 6) The relevant explanations for the dissipation decrease (stiffness increase) are described in detail in PMID: 26269427, which we now cite for the interested reader. Therefore, increased HS film stiffness by QCM-D and reduced HS diffusion by FRAP, as shown in our work, provides solid evidence for cross-linking. Please also see the explanations and added data in our revised manuscript (Supplementary Fig. 6).

10) The authors claim that Shh cross-link two HS chains via its two independent HS-binding sites and “relay” between HS chains. They also claim that this “relay” contributes to obtaining the distribution “speed” of Hhs. This reviewer thinks this is an over-assumption that is not based on the experimental evidence. Indeed, their FRAP indicates that Shh with complete HS binding site, which supposes to be “relayed” between HS chains, moves more slowly than the mutated Shh that is supposed not to be relayed. Authors need to provide experimental data and a clear explanation supporting their proposal.

We note that the link to FRAP is misleading, as the FRAP assay assessed HS mobility and not Hh mobility. Therefore, there is no contradiction – both assays support cross-linking of HS by Shh. We would also like to point out that our QCM-D data showing Shh-induced HS film rigidification, together with immediate Shh detachment from the film upon the addition of soluble heparin, and in combination with the reduced abilities of both Shh R/A and R/E mutants to do so (Fig. 6), and in combination of no observable switching to lower-sulfated HS forms (Fig. 7a) jointly provide clear evidences for inter-chain cross-linking by Shh and relay between the chains.

Minor points:

1. Specific terms that first appear is needed to be defined. e.g., pdb in P5, L106.

We now define all used abbreviations in the manuscript.

2. It is not clear why three independent data for one genotype are shown only in Fig 2E. The authors need to explain this.

We have changed the figure and pooled the data from the three previously shown injection lines into one data set.

3. In Fig. 3 A and B, the labels of the first column in each panel are lacking.

We corrected this mistake.

4. For mosaic analysis of wing imaginal disc in Fig.4, authors need to mention or show data about the cell competition, since they mention it in the eye-imaginal disc case.

We added this information to the revised manuscript (Supplementary Fig. 5). Supplementary movie 1 was added to the revision to demonstrate that the entire disc is formed from hh allele bearing twin spots. GFP signals only stem from GFP-positive heterozygous glia that migrate from the brain through the optic stalk and some superficial cells located on top of the disc.

5. The preparation of de-6O- and de-2O- soluble heparin has not appeared in the material and methods. If the authors prepare those materials in their hands, the evaluation of prepared de-6O and de-2O-soluble heparin is needed to be presented.

The selectively desulfated heparins were purchased from Iduron (Manchester, UK) and also characterized by the company. The catalogue numbers are: N-Desulfated Heparin DSH003/N, 6-O-Desulfated Heparin DSH002/6, and 2-O-Desulfated Heparin DSH001/2. This is now stated in the methods and materials section.

REVIEWERS' COMMENTS

Reviewer #1 (Remarks to the Author):

Authors included the required additional information.

Reviewer #2 (Remarks to the Author):

The points raised in review were sufficiently addressed, and there are a few points arising in the re-submission:

The new Fig 1F is confusing to me. I believe that this is the same underlying data as Fig 1F from the original submission, but that graph was much more intuitive to read, rather than showing the posterior on the estimated B values. consider reverting this change.

The new Fig 1G is unclear. This figure tries to make the point that 1) Ptc is not saturated over the concentration ranges the authors are working with and 2) that the alleles do not affect the Ptc sensitivity. the data should be plotted quantitatively, i.e. the x axis should be concentration of Shh, and the y axis should be response (A405). there should be 3 lines showing the response as a function of Shh, perhaps color-coded by Shh allele. The bevy of pairwise statistical comparisons is unnecessary and distracting. Also the experiment is technically very simple to perform on a plate reader, and the figure would be stronger if it included a wider range of Shh concentrations to sample fully un-induced and saturating levels over a ~8-step ~100 or 1000-fold concentration range. the data they have presented is minimal, although it is sufficient to support their claims. it is notably below the standard for cleanliness that the rest of the manuscript adheres to.

Reviewer #3 (Remarks to the Author):

This reviewer found the revised manuscript improved, and I was nearly convinced of the author's claim. The minor points for the current manuscript are below.

In supplement figure 5b, authors should point out where the developing ocelli in hh[KO] and hh[KO;hhR/E] eye discs are. Also, it is better to indicate where the MFs are in the eye discs of every genotype so that the readers can easily recognize that the expression of CI155 is less affected in the hh[KO;hhR/E] eye disc. For those points, it is better to provide a magnified image of developing ocelli and MFs of the above genotype, as the authors do so in the case of developing ocelli of hh[KO;hh].

NCOMMS-22-25488A Reply to reviewer's comments:

Reviewer #2 (Remarks to the Author):

The new Fig 1F is confusing to me. I believe that this is the same underlying data as Fig 1F from the original submission, but that graph was much more intuitive to read, rather than showing the posterior on the estimated B values. consider reverting this change.

Reply: we changed this figure into a form that is now much more intuitive to understand. The source data are indeed the same as shown in the initial submission.

The new Fig 1G is unclear. This figure tries to make the point that 1) Ptc is not saturated over the concentration ranges the authors are working with and 2) that the alleles do not affect the Ptc sensitivity. the data should be plotted quantitatively, i.e. the x axis should be concentration of Shh, and the y axis should be response (A405). there should be 3 lines showing the response as a function of Shh, perhaps color-coded by Shh allele. The bevy of pairwise statistical comparisons is unnecessary and distracting. Also the experiment is technically very simple to perform on a plate reader, and the figure would be stronger if it included a wider range of Shh concentrations to sample fully un-induced and saturating levels over a ~8-step ~100 or 1000-fold concentration range. the data they have presented is minimal, although it is sufficient to support their claims. it is notably below the standard for cleanliness that the rest of the manuscript adheres to.

Reply: We have changed the figure according to this reviewers suggestion.

Reviewer #3 (Remarks to the Author):

This reviewer found the revised manuscript improved, and I was nearly convinced of the author's claim. The minor points for the current manuscript are below.

In supplement figure 5b, authors should point out where the developing ocelli in hh[KO] and hh[KO;hhR/E] eye discs are. Also, it is better to indicate where the MFs are in the eye discs of every genotype so that the readers can easily recognize that the expression of Cl155 is less affected in the hh[KO;hhR/E] eye disc. For those points, it is better to provide a magnified image of developing ocelli and MFs of the above genotype, as the authors do so in the case of developing ocelli of hh[KO;hh].

Reply: We have now added the required information to the supplementary figure.